# *Lactobacillus* supports Clostridiales to restrict gut colonization by multidrug-resistant *Enterobacteriaceae*

Ana Djukovic [1,2], María José Garzón [1], Cécile Canlet[3,14], Vitor Cabral[4,14], Rym Lalaoui[5,14], Marc García-Garcerá [6,14], Julia Rechenberger [7,14], Marie Tremblay-Franco[3], Iván Peñaranda[1], Leonor Puchades-Carrasco [8], Antonio Pineda-Lucena[8,9], Eva María González-Barberá[10], Miguel Salavert[10], José Luis López-Hontangas[10], Miguel Á. Sanz [11], Jaime Sanz[10,12], Bernhard Kuster [7], Jean-Marc Rolain[5], Laurent Debrauwer[3], Karina B. Xavier [4], Joao B. Xavier [2] & Carles Ubeda [1,13] ✉

Infections by multidrug-resistant *Enterobacteriaceae* (MRE) are life-threatening to patients. The intestinal microbiome protects against MRE colonization, but antibiotics cause collateral damage to commensals and open the way to colonization and subsequent infection. Despite the significance of this problem, the specific commensals and mechanisms that restrict MRE colonization remain largely unknown. Here, by performing a multi-omic pro-spective study of hospitalized patients combined with mice experiments, we find that *Lactobacillus* is key, though not sufficient, to restrict MRE gut colo-nization. *Lactobacillus rhamnosus* and *murinus* increase the levels of Clos-tridiales bacteria, which induces a hostile environment for MRE growth through increased butyrate levels and reduced nutrient sources. This mechanism of colonization resistance, an interaction between *Lactobacillus* spp. and Clostridiales involving cooperation between microbiota members, is conserved in mice and patients. These results stress the importance of exploiting microbiome interactions for developing effective probiotics that prevent infections in hospitalized patients.

Multidrug-resistant *Enterobacteriaceae* (MRE), including *Escherichia coli* and *Klebsiella pneumoniae*, have become a major threat for patient health[1]. These pathogens have acquired resistance to the majority of available antibiotics, which complicates their treatment. MRE are considered the third most critical resistant bacterial pathogens for which new therapies should be developed[2].

MRE infections frequently start with intestinal colonization[3]. The gut microbiota can protect its host against MRE, a phenomenon known as "colonization resistance"[4]. Antibiotic treatment, however, damages the intestinal microbiota of patients and enables intestinal coloniza-tion by MRE to extremely high levels[5]. Once at high intestinal density,

MRE can disseminate to the bloodstream and cause life-threatening infection[6]. High density of resistant pathogens in one patient also promotes dissemination to other patients through fecal environ-mental contamination[7,8]. Therefore, understanding how the micro-biome protects against MRE colonization and intestinal expansion is crucial to curb infections among hospitalized patients. Understanding the mechanisms through which commensal bacteria confer protection can lead to novel microbiome-based approaches to restrict MRE colonization and subsequent infections.

Recent studies have sought to characterize the microbiome of patients colonized with antibiotic-resistant *Enterobacteriaceae* to

identify commensals involved in colonization resistance[9,10]. Although clinical studies have identified potentially protective commensal microbes[9,10], candidates were not tested experimentally and mechanisms of protection against MRE colonization remained to be elucidated.

Mouse studies, on the other hand, have provided potential mechanisms by which the microbiota confers protection against *Enterobacteriaceae* pathogens[11], including production of inhibitory molecules, competition for nutrients or immune system induction[12–15]. However, not all mechanisms found in animal models are present in humans as well[16]. Thus approaches that combine the analysis of human and mouse data are needed for elucidating relevant mechanisms for MRE colonization resistance.

Here we applied multiple omic techniques to study the gut microbiome of hospitalized patients, specific-pathogen-free (SPF) mice and gnotobiotic mice to identify protective commensal microbes and strategies such microbes use to provide colonization resistance against MRE. We found that *Lactobacillus sp.* is associated with resistance to MRE intestinal colonization in hospitalized patients. Next, using a mouse model we demonstrated that *Lactobacillus sp.* is required but not sufficient to restrict MRE intestinal colonization (i.e. two orders of magnitude lower MRE levels compared to control mice). We demonstrate in mice that one of the mechanisms by which *Lactobacillus sp.* diminishes MRE colonization is by promoting the expansion of Clostridiales genera. This cooperation among phylogenetically distant commensals results in a hostile microenvironment for MRE, consisting of higher levels of the inhibitory metabolite butyrate and decreased levels of key nutrients for MRE growth. Multi-omic analysis of hospitalized patients revealed the same association between *Lactobacillus* and Clostridiales in humans, which correlates negatively with MRE gut colonization. Our results highlight the importance of interactions between commensal bacteria for the design of microbiome-based therapies to prevent infections by antibiotic-resistant pathogens.

## Results

### *Lactobacillus* is associated with reduced multidrug-resistant *Enterobacteriaeae* colonization in patients with acute leukemia

We previously collected weekly fecal samples ($N = 802$) from hospitalized acute leukemia patients (ALP; $N = 133$) to identify clinical variables associated with week-to-week changes in the levels of MRE[17]. ALP often receive antibiotics that potentially disrupt their microbiota and lead to frequent colonization by MRE. Therefore, ALP represent an optimal population to study how changes in the microbiota influence MRE intestinal colonization in humans. We reported that 60% of ALP from our cohort harbor detectable levels of MRE[17]. Moreover, we found that fecal MRE levels change considerably between different patients and within the same patient over time[17]. An example of these changes is reflected in Fig. 1a, which shows the dynamics of the MRE levels detected in one particular patient that was colonized during the hospitalization period and for which multiple timepoints could be analyzed (Fig. 1a). Besides the MRE levels, we also determined the antibiotic resistance pattern and taxonomy of the isolated MRE (Fig. 1a). In addition, we acquired clinical data that could be relevant for explaining MRE intestinal colonization dynamics, such as the antibiotics received by the patient (Fig. 1a) and other variables previously described (e.g. admission type, comorbidities, etc.)[17]. Here, we aim to identify changes in the microbiome associated with MRE intestinal colonization levels, which could allow us to identify commensal bacteria that are key to confer resistance to these multidrug-resistant pathogens in humans. To this end, we analyzed the microbiota of ALP through 16S rRNA sequencing. We included patients in which MRE colonization was detected in at least one sample and for which we were able to collect at least a follow-up sample during the same hospital admission period so that we could track MRE and microbiome changes

overtime (Supplementary Data File 1). A total of 179 samples from 47 patients were included in the study after excluding specific samples to avoid cofounding clinical variables that could hamper the identification of interactions between the microbiome and MRE (see methods). We applied a dynamic approach, using longitudinal microbiome and MRE data, to test the hypothesis that changes in MRE levels are associated with changes in the microbiota composition. To this end, we calculated the log2 Fold Change (log2FC) of the abundance of each bacterial taxon (Supplementary Data File 2; genus level) between pairs of consecutive samples collected from the same patient and studied how these correlate with changes (log2FC) in MRE levels. Principal coordinate of analysis (PCoA) using the log2FC data showed that microbiota changes were significantly associated with changes in MRE levels (Fig. 1b, $p = 0.017$, $R^2 = 0.018$, PERMANOVA). A similar result, albeit less significant, was obtained when this analysis was performed at the operational taxonomical units (OTU) level ($p = 0.046$, $R^2 = 0.014$, PERMANOVA). To identify the bacterial taxa, whose changes were associated with changes in MRE levels, we correlated the log2FC of the abundance of each bacterial taxon with the log2FC in MRE levels. We focused our analysis on the most prevalent/abundant taxa (median > 0.01%) that could be classified at least to the family level (Fig. 1a). Notably, the analysis identified two genera whose change over time was significantly associated with a change in MRE levels (Two-sided Spearman correlation test and Benjamini–Hockberg correction, $p < 0.01$; $q < 0.05$, the exact $p$ and $q$ values are shown in Supplementary Data File 3). Changes in the genus *Parabacteroides* were positively associated with changes in MRE levels (Fig. 1c, Supplementary Fig. 1, $\rho = 0.267$, $p = 0.006$). In contrast, changes in the abundance of *Lactobacillus* were negatively associated with MRE dynamics (Fig. 1c, d; two-sided Spearman correlation test, $\rho = -0.274$, $p = 0.004$). In other words, an increase in the abundance of *Lactobacillus* was associated with a decrease in MRE levels, pinpointing *Lactobacillus* as a potential protective commensal against MRE intestinal colonization. A similar analysis was performed with less prevalent/abundant bacteria (median < 0.01%), but none of the associations remained significant after adjusting for multiple comparisons (two-sided Spearman correlation test and Benjamini–Hockberg correction, $q > 0.1$, the exact $q$ values are shown in Supplementary Data File 3).

To confirm the negative interaction between *Lactobacillus* and MRE and discard the possibility that this interaction was mainly driven by a few patients that were oversampled, we applied linear mixed effects model to simultaneously analyze the variance of *Lactobacillus* abundance as a fixed effect and the inter-patient variability as a random effect. A similar negative relationship between the dynamics of *Lactobacillus* and MRE was detected using this analysis (lm.coeff = −0.447, $p = 0.023$). Consistent with this negative interaction, we also detected that fecal samples with higher levels of *Lactobacillus* (>1% abundance) contain significantly lower levels of MRE (Fig. 1e; two-sided Mann–Whitney test, $p = 0.006$).

Considering the potential negative impact that *Lactobacillus* could have on MRE levels, we analyzed if this was due to any specific *Lactobacillus* OTU. The OTU with the most significant association with MRE levels was the OTU20 (Supplementary Data File 3), although the association was not as strong as with the whole genus (Supplementary Fig. 2, Spearman one-sided test, $\rho = -0.187$, $p = 0.027$). In addition, this OTU, being the most prevalent one within this genus (Supplementary Data File 3), was present in 35% of the samples (68% of patients) as compared to the genus *Lactobacillus* that is present in 72% of the samples (93% of patients), suggesting that the negative association between *Lactobacillus* and MRE might be due to more than one OTU.

In summary, the microbiota analyses from our patients' cohort indicate that MRE gut dynamics in patients are associated with specific microbiota changes and pinpoint a potential antagonistic effect of the genus *Lactobacillus* on MRE intestinal colonization.

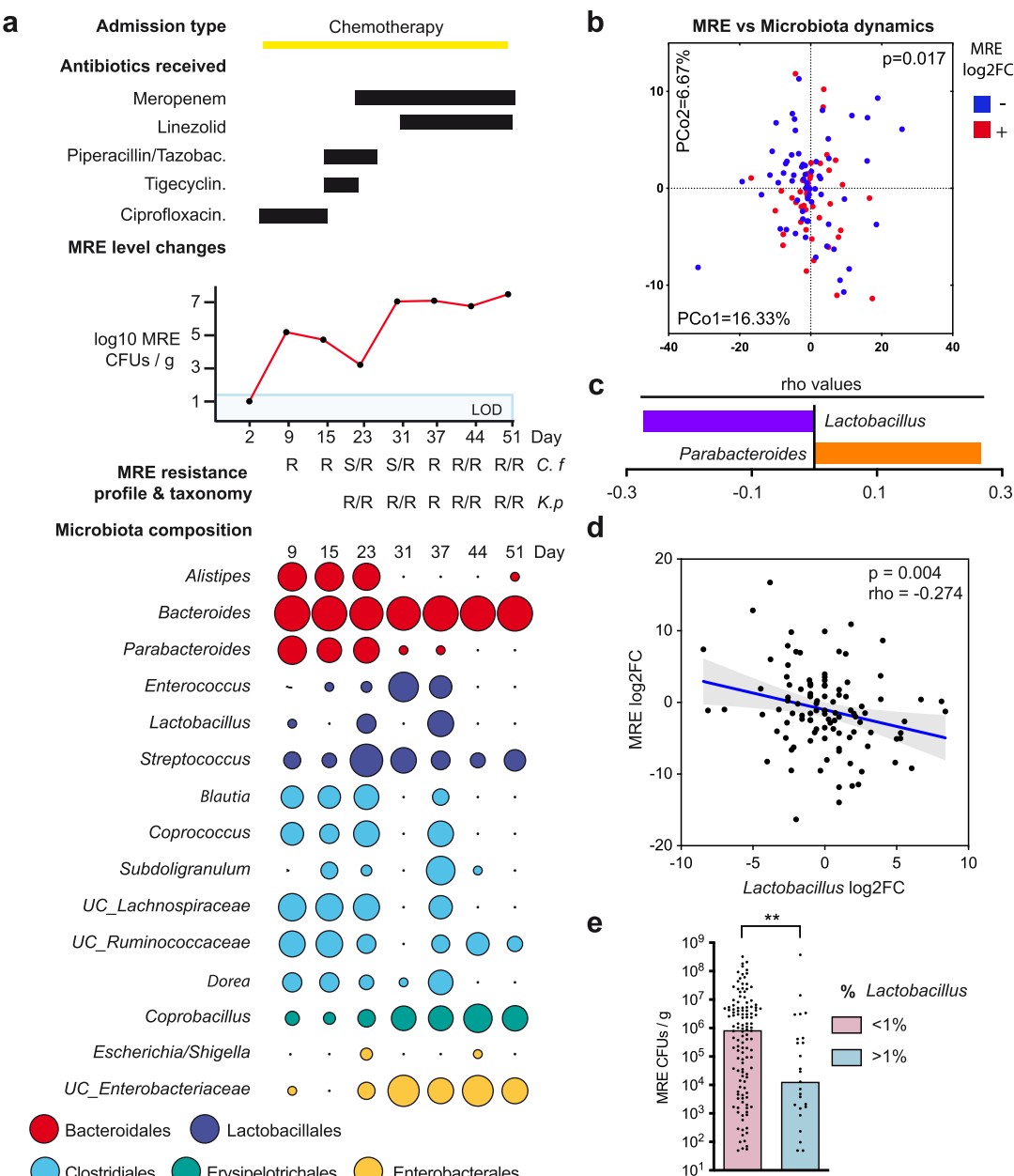

**Fig. 1 | *Lactobacillus* is associated with reduced multidrug-resistant *Enterobacteriaceae* (MRE) colonization in patients with acute leukemia (ALP). a** For each patient we collected clinical data (e.g. antibiotic treatment, type of hospital admission) as described in our previous publication[17]. Fecal samples were collected every week. From each sample, we quantified the MRE levels and characterized their taxonomy and antibiotic resistance[17]. Microbiome data was obtained from each sample, starting the first-day MRE was detected. The figure shows the data obtained from one of the patients at different timepoints (days of hospitalization). The levels of MRE are shown in log10 scale. LOD limit of detection. C.f *Citrobacter freundii*, K.p *Klebsiella pneumoniae*. R/S isolate resistant or sensitive to the received antibiotics. The microbiota composition reflects the abundance of the most prevalent taxa (median > 0.01%). The size of the circle is proportional to the abundance of the taxa. Colors indicate the bacterial order that the taxa belong to. UC unclassified. **b** Principal coordinate of analysis (PCoA) obtained with Euclidean distances between pairs of samples using a matrix containing the log2 fold change (log2FC) in relative abundances for identified taxa (genus level) in a pair of consecutive samples collected from a patient. PCoA shows that the change in MRE levels is associated with a change in the microbiota composition. PERMANOVA, $p = 0.017$, $N = 106$ pairs of samples. **c** Taxa whose change in abundance was associated with a change in MRE levels. Two-sided Spearman correlation test, $p < 0.01$ and Benjamini–Hockberg correction ($q < 0.05$), $N = 106$ pairs of samples. Bars represent the rho values. **d** Log2FC in the levels of *Lactobacillus* as compared to the log2FC in the levels of MRE in pairs of consecutive samples collected from ALP. Two-sided Spearman correlation test, $p = 0.004$, $N = 106$ pairs of samples. The line represents the linear regression mean and the gray shadow the 95% CI. **e** MRE-colonized samples from ALP with higher abundances of the genus *Lactobacillus* (>1%) had significantly lower MRE colonization levels as compared to samples that had lower *Lactobacillus* levels (<1%). Two-sided Mann–Whitney test, **$p = 0.006$, $N = 126$ and 25 samples per group. Bars represent de median. Source data are provided as a Source Data file.

## *Lactobacillus spp.* is required but not sufficient to restrict multidrug-resistant *Enterobacteriaceae* gut colonization in mice

In order to demonstrate the protective role of *Lactobacillus* against MRE intestinal colonization, we first isolated a representative of the most prevalent *Lactobacillus* OTU from patient samples (see methodology for isolation in the "Methods" section). This OTU is also present in other cohorts of healthy individuals of different continents (Supplementary Fig. 3). Whole-genome sequencing

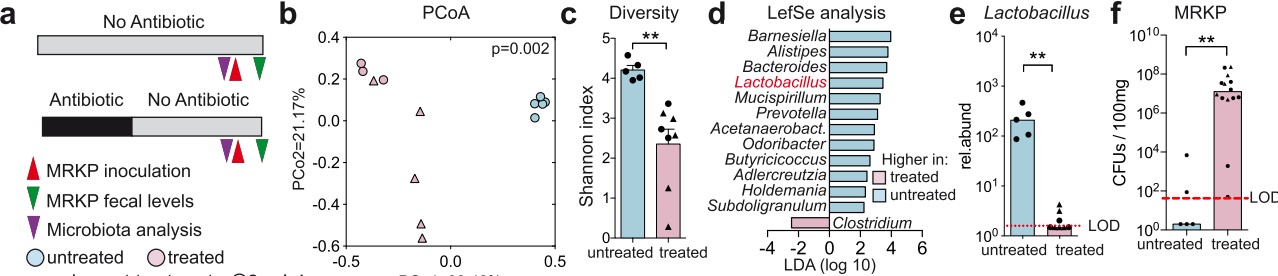

**Fig. 2 | Antibiotic treatment depletes *Lactobacillus* and promotes gut colonization by multidrug-resistant *K. pneumoniae* (MRKP) in mice. a** Schematic representation of the experimental model. Mice were treated with antibiotics (AVN) for 1 week or left untreated. Two or 4 weeks (w) after antibiotic cessation fecal samples were collected for microbiota analysis and, subsequently, mice were orally gavaged with MRKP. **b** Principal coordinate of analysis (PCoA) of the Bray–Curtis distances at the OTU level. The microbiota composition of untreated animals is significantly different from the microbiota of mice that were left to recover after antibiotic cessation. PERMANOVA, $p = 0.002$, $N = 5$ and 8 mice per group. **c** Mice that were allowed to recover after antibiotic cessation do not recover the bacterial diversity levels of untreated mice. Diversity was calculated applying the Shannon index at the OTU level. Two-sided *T*-test, **$p = 0.003$, $N = 5$ and 8 mice per group. **d** LEfSe analysis was performed to identify the most discriminative microbiota changes between samples collected from untreated mice and samples collected 2

or 4 weeks after stopping AVN treatment ($p < 0.05$, Linear discriminant analysis (LDA) > |2|). LDA score, $p$ values, and bacteria that could not be classified up to the genus level are shown in Supplementary Data File 5. $N = 5$ and 8 mice per group. **e** 16S rRNA-based relative abundance (rel.abund) of *Lactobacillus* (counts per 10000 sequences) in mice that recovered for 2 or 4 weeks after AVN treatment as compared to mice that did not receive any treatment. Two-sided Mann–Whitney test, **$p = 0.002$, $N = 5$ and 8 mice per group. LOD limit of detection. **f** MRKP colonization levels on the first-day post inoculation were significantly lower in untreated mice as compared to animals that were allowed to recover for 2 or 4 weeks after antibiotic treatment before the MRKP inoculation. Two-sided Mann–Whitney test, **$p = 0.001$, $N = 5$ and 14 mice per group. CFUs colony-forming units. Bars represent the average in **c**, the LDA score in **d** and the median in **e**, **f**. Whiskers represent the SEM in **c**. Source data are provided as a Source Data file.

(WGS) confirmed that the isolate belongs to the genus *Lactobacillus*, more precisely species *Lactobacillus rhamnosus* (now renamed to *Lacticaseibacillus rhamnosus*[18]) (Supplementary Fig. 4; 98.7% identity). Extended analysis of our isolate that involved comparison with *L. rhamnosus* genomes deposited at NCBI ($N = 198$), including some used as probiotics, indicated that none of the sequenced strains is 100% identical to the strain isolated in our study (Supplementary Fig. 5; Supplementary Data File 4). Next, we established a mouse model that would allow us to demonstrate the role of *Lactobacillus* in conferring protection against MRE intestinal colonization. We and others have previously shown that antibiotic treatment in mice produces permanent changes in the microbiome that allow intestinal colonization by opportunistic pathogens, including vancomycin-resistant *Enterococcus* (VRE) or *Clostridoides difficile*[19–21]. Precise restoration of the microbiome through inoculation of bacterial isolates restricts pathogen intestinal colonization[22]. Based on these results, we set up a mouse model in which the microbiota of C57BL/6 mice was disrupted with a cocktail of antibiotics containing ampicillin, vancomycin and neomycin (AVN) during 1 week (Fig. 2a). Subsequently, antibiotic treatment was stopped and mice were allowed to recover for 2 or 4 weeks. Taking into account our previous published data[19], AVN treatment should permanently deplete major members of the microbiota, including the genus *Lactobacillus*. After the recovery period, the murine microbiota was analyzed and mice were inoculated with an MRE strain from the species *K. pneumoniae* (MRKP), which was previously used to study mechanisms of colonization resistance against MRE[12]. Untreated mice gavaged with MRKP served as a control group. As predicted, AVN produced permanent changes in the microbiota (Fig. 2b–e; PERMANOVA, $p = 0.002$, $R^2 = 0.369$). These included the depletion of *Lactobacillus* (Fig. 2e, two-sided Mann–Whitney test, $p = 0.002$), which was one of the most significant changes that discriminated AVN-treated mice from untreated mice (Fig. 2d; LEfSe, $p < 0.05$ and Linear discriminant analysis (LDA) > |2|, exact $p$ and LDA values are shown in Supplementary Data File 5). Notably, these microbiota changes allowed high levels of MRKP intestinal colonization (Fig. 2f, two-sided Mann–Whitney test, $p = 0.001$). In contrast, 24 h after MRKP inoculation most control mice had no detectable MRKP (Fig. 2f). In

concordance with the patient data, the levels of *Lactobacillus* pre-MRKP inoculation negatively correlated with the MRKP colonization capacity (two-sided Spearman test, $p = 0.007$, $\rho = -0.69$).

Next, we used this mouse model to demonstrate the inhibitory effect of *Lactobacillus* on MRKP colonization. After AVN treatment, the *L. rhamnosus* strain isolated from patients was administered to mice and 2 weeks later MRKP was inoculated. As expected, significantly higher levels of *Lactobacillus* were detected in mice that received *L. rhamnosus* as compared to AVN-treated animals that did not receive it (Fig. 3b, two-sided Mann–Whitney test, $p = 3e-5$). Most importantly, *L. rhamnosus* administration significantly reduced the capacity of MRKP to colonize the murine gut (Fig. 3c, Supplementary Fig. 6). Significantly lower MRKP levels were detected in feces (Fig. 3c, Supplementary Fig. 6, Two-sided Mann–Whitney test, $p = 0.0003$, $p = 0.045$) and the colonic mucosa (Supplementary Fig. 6, two-sided Mann–Whitney test, $p = 0.004$) 1 or 2 days after MRKP inoculation in the group of mice that received *L. rhamnosus*.

We also isolated a *Lactobacillus* strain from untreated mice to test if the protective effect of a *Lactobacillus* strain more adapted to the mouse gut could confer a higher level of protection. WGS confirmed that this additional isolate belongs to the genus *Lactobacillus* and revealed 99.3% identity to the species *L. murinus* (Supplementary Fig. 4), a species often found in mouse colonies[23,24]. *L. murinus* achieved the same level of gut colonization (Fig. 3b) and restricted MRKP intestinal colonization to a similar extent as *L. rhamnosus* (Fig. 3c), indicating that at least two different species of *Lactobacillus* can restrict MRKP intestinal colonization.

Besides *Lactobacillus*, we also tested the protective capacity of another murine commensal that was depleted after AVN treatment and thus might be associated with protection against MRKP colonization (i.e. *Barnesiella*, Fig. 2d). However, in contrast to *Lactobacillus*, mice that received *Barnesiella* were equally susceptible to gut colonization as mice that did not receive any bacteria (Supplementary Fig. 7). Moreover, administration of *Barnesiella* in combination with *Lactobacillus* did not enhance the capacity of *Lactobacillus* to restrict MRKP colonization (Supplementary Fig. 7C), despite both commensals' ability to efficiently co-colonize the murine gut (Supplementary Fig. 7D). These results further

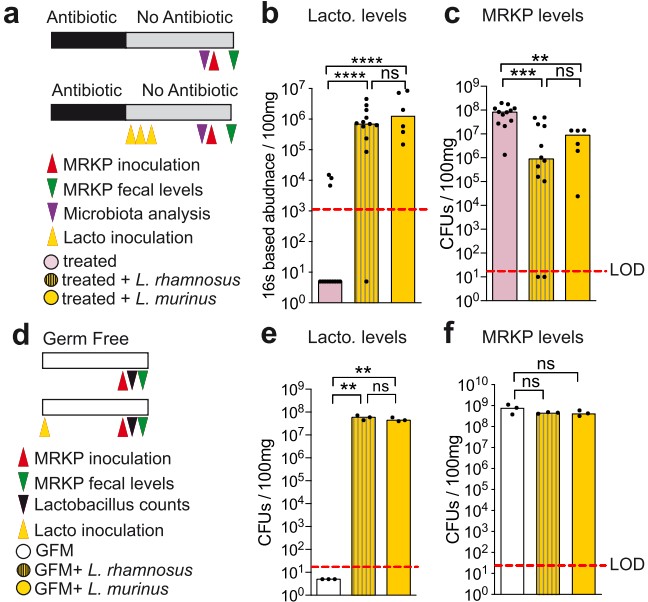

**Fig. 3 | *Lactobacillus spp.* is required but is not sufficient to restrict multidrug-resistant *K. pneumoniae* (MRKP) gut colonization in mice. a** Schematic representation of the experimental model. Mice were treated with antibiotics (AVN) for 1 week in the drinking water. A group of mice received an isolate *L. murinus* or *L. rhamnosus* during 3 consecutive days by oral gavage, starting one day after antibiotic cessation. A control group of mice received the bacterial vehicle instead (PBS-glycerol cysteine). Two weeks post antibiotic cessation, fecal samples were collected for microbiota analysis and subsequently mice were orally gavaged with MRKP. **b** Levels of *Lactobacillus* pre-MRKP inoculation in each group of mice shown in (**a**) calculated by normalizing the relative abundances obtained through 16S rRNA sequencing with the total bacterial fecal numbers inferred from 16 S rRNA qPCR data. Two-sided Mann–Whitney test, ****$p < 0.0001$, ns not significant: $p = 0.553$, $N = 12$, $N = 12$ and $N = 6$ mice per group. LOD limit of detection. **c** Levels of MRKP in feces one day post MRKP inoculation in each group of mice shown in (**a**). Two-sided Mann–Whitney test, ***$p = 0.0003$, **$p = 0.002$, ns not significant: $p = 0.664$, $N = 12$, $N = 12$ and $N = 6$ mice per group. CFUs: colony-forming units. **d** Schematic representation of the experimental model. Germ-free mice (GFM) were inoculated with *L. murinus*, *L. rhamnosus* or left germ free. Ten days after, mice were orally gavaged with MRKP. **e** Levels of *Lactobacillus* quantified by plating fecal samples from each group of mice shown in (**d**) in selective media. Two-sided *T*-test, **$p < 0.01$, ns not significant: $p = 0.325$, $N = 3$ mice per group. **f** Levels of MRKP in feces one day post MRKP inoculation in each group of mice shown in (**d**). Two-sided Mann–Whitney test, ns not significant: $p > 0.05$, $N = 3$ mice per group. Bars represent the median. Lacto: *Lactobacillus*. Source data are provided as a Source Data file.

support a specific role of *Lactobacillus* in reducing MRKP gut colonization levels.

Having shown that *Lactobacillus* can restrict MRKP intestinal colonization in antibiotic-treated mice, we next evaluated the capacity of *Lactobacillus* to restrict MRKP gut colonization in Germ-Free Mice (GFM). As opposed to AVN-treated mice, GFM do not contain any residual microbiota, which would allow us to test if *Lactobacillus* is sufficient to confer protection. We colonized GFM with either *L. murinus* or *L. rhamnosus* for 10 days before MRKP inoculation (Fig. 3d). We compared MRKP colonization levels in *Lactobacillus* monocolonized mice with uncolonized germ-free controls. As expected, control mice were highly colonized with MRKP (Fig. 3f). However, GFM monocolonized with either *L. murinus* or *L. rhamnosus* did not restore colonization resistance: the levels of MRKP were similar to those observed in controls (Fig. 3f) despite efficient gut colonization of GFM by *Lactobacillus* strains (Fig. 3e).

## *Lactobacillus* promote the recovery of Clostridiales genera, which restricts multidrug-resistant *K. pneumoniae* intestinal colonization in mice

The results obtained in GFM indicated that gut colonization by *Lactobacillus* alone is insufficient to restrict MRKP colonization. One possible explanation is that *Lactobacillus* do not suppress MRKP growth directly, but rather promote the recovery of other commensal bacteria that diminish MRKP growth in antibiotic-treated mice. To investigate this, we analyzed the microbiota of antibiotic-treated mice that had received either *L. murinus* or *L. rhamnosus* and compared it with mice that did not receive any *Lactobacillus* strain. Mice that received the *Lactobacillus* strains had a significantly more diverse residual microbiota 2 weeks after antibiotic cessation (Fig. 4a, two-sided *T*-test, $p = 0.004$). This increase in diversity was associated with a better recovery of specific genera, mainly from the order Clostridiales (Fig. 4b). We detected six genera whose abundances significantly increased in mice that received *Lactobacillus* strains (Fig. 4b, two-sided Ancom2 test and Benjamini–Hockberg correction, $p < 0.05$, $q < 0.1$, the exact $p$ and $q$ values are shown in Supplementary Data File 6). Four out of the 6 identified genera belong to the order Clostridiales−*Pseudoflavonifractor*, *Flavonifractor*, *Oscillibacter*, and *Lachnoanaerobaculum*. Notably, the levels of these commensal genera increased to a similar extent in both mice that received *L. murinus* and *L. rhamnosus* (Supplementary Fig. 8), indicating that two different *Lactobacillus* species can have a similar impact on the expansion of these commensal Clostridiales. The two other genera whose abundances significantly increased upon *Lactobacillus* administration were *Allobaculum* (Erysipelotrichales order) and, as expected, *Lactobacillus* (Fig. 4b). In addition, we detected an increase in abundance of bacteria that could not be classified to the genus level but belonged to the family *Lactobacillaceae* and the Clostridiales family *Ruminococcaceae* (Supplementary Data File 6). In contrast, *Lactobacillus* administration had no significant effect on the abundances of other 14 analyzed taxa that passed our threshold criteria (median > 0.01% in any of the two groups) (Supplementary Data File 6). Interestingly, changes in the microbiota composition induced by *Lactobacillus* were associated with a moderate, albeit significant increase in the overall density of the microbiota, based on qPCR of the 16S rRNA gene (Supplementary Fig. 9A, two-sided Mann–Whitney test, $p = 0.012$). Moreover, when the relative abundances obtained through 16S rRNA sequencing were normalized with the total number of 16S rRNA gene copies per sample, a significant increase in the absolute levels of seven Clostridiales genera (*Pseudoflavonifractor*, *Flavonifractor*, Oscillibacter, *Ruminococcus2*, *Lachnoanaerobaculum*, *Marvinbryantia,* and *Moryella*) and other three genera (*Lactobacillus*, *Allobaculum,* and *Akkermansia*) was detected in mice that received *Lactobacillus* (Supplementary Fig. 9B, two-sided Mann–Whitney test and Benjamini–Hockberg correction, $p < 0.05$, $q < 0.1$, the exact $p$ and $q$ values are shown in Supplementary Data File 7). Consistent with a potential direct effect of *Lactobacillus* in promoting the growth of certain microbiome taxa, products derived from *Lactobacillus* metabolism significantly boosted the in vitro growth of *Flavonifractor*, one of the Clostridiales genera that expanded upon *Lactobacillus* administration (see methods, Supplementary Fig. 10). Altogether these results suggest that *Lactobacillus* administration promotes the expansion of specific members of the microbiota, the majority of which belong to the Clostridiales order.

Several studies in mice have shown that Clostridiales play a role in inhibiting *Enterobacteriaceae* strains that are sensitive to antibiotics[25,26]. Thus, we hypothesized that in our model *Lactobacillus* administration could restrict MRKP intestinal colonization by promoting the expansion of Clostridiales taxa. In support of this hypothesis, we found that abundances of commensal bacteria from order Clostridiales pre-infection had the strongest negative association with MRKP fecal levels 1 day post inoculation (Fig. 4c, Two-sided

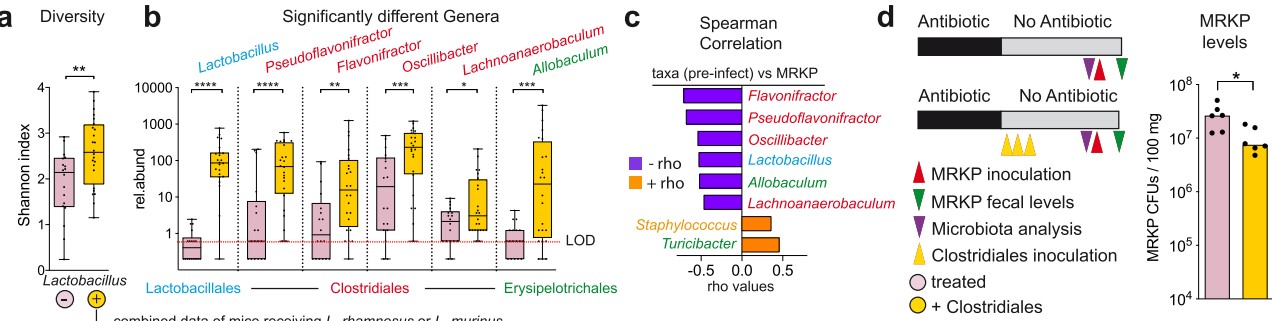

**Fig. 4 | *Lactobacillus* promote the expansion of Clostridiales commensals which restricts multidrug-resistant *K. pneumoniae* (MRKP) gut colonization in mice.**
**a** Bacterial diversity is partially recovered after administration of *Lactobacillus* strains (*L. murinus* or *L. rhamnosus*, combined data) to antibiotic (AVN)-treated mice. Shannon index is significantly higher in animals that received *Lactobacillus*, two-sided *T*-test, **$p = 0.004$, $N = 18$ and 24 mice per group. **b** Statistically significant differences in levels of different microbiota members were detected upon *Lactobacillus* administration. Two-sided Ancom2 test,****$p < 0.0001$, ***$p < 0.001$, **$p = 0.002$, *$p = 0.028$ and Benjamini–Hockberg correction ($q < 0.1$), $N = 18$ and 24 mice per group. Only bacteria that could be classified up to the genus level are shown. *p* and *q* values for all analyzed genera and other bacteria that could not be classified to the genus level are shown in Supplementary Data File 6. Rel.abund: relative abundance: counts per 10000 sequences. LOD limit of detection.
**c** Correlation analysis between the levels of commensal bacteria (pre-MRKP inoculation, pre-infect) with the levels of MRKP in feces 24 h after MRKP inoculation. Two-sided Spearman correlation test & Benjamini–Hockberg correction,

$p < 0.05$, $q < 0.05$, $N = 42$ mice. Only significant bacteria that could be classified up to the genus level are shown. $p/\rho$ values for all analyzed genera and other bacteria that could not be classified at the genus level are shown in Supplementary Data File 8. Bars represent the Spearman correlation rho values. **d** Schematic representation of the experimental model. Mice were treated with antibiotics (AVN) for 1 week in the drinking water or left untreated. A group of mice received a cocktail containing four Clostridiales strains during three consecutive days. Another group received the bacterial vehicle instead. Two weeks post antibiotic cessation mice were orally gavaged with MRKP. Fecal levels of MRKP one day after inoculation show that administration of Clostridiales significantly decreases the MRKP colonization levels. Two-sided Mann–Whitney test, *$p = 0.026$, $N = 6$ mice per group. CFUs colony-forming units. Boxes extend from the 25th to 75th percentiles. The line within the boxes represents the median. Whiskers indicate the maximum and minimum values. Bars in D represent the median. Source data are provided as a Source Data file.

Spearman Correlation and Benjamini–Hockberg correction, $p < 0.01$, $q < 0.01$, the exact p and q values are shown in Supplementary Data File 8). This negative association was even stronger than the one detected between *Lactobacillus* and MRKP levels (Fig. 4c). We next evaluated if a higher recovery of Clostridiales could restrict MRKP growth, independently from the presence of *Lactobacillus*. To this end, we administered a cocktail of four Clostridiales isolates to antibiotic-treated mice (see "Methods"). Two weeks after antibiotic cessation, we detected higher levels of the administered Clostridiales in mice receiving the cocktail (Supplementary Fig. 11, Two-sided Ancom2 test and Benjamini–Hockberg correction, $p < 0.01$, $q < 0.05$, the exact p and q values are shown in Supplementary Data File 9). As expected, the levels of *Lactobacillus* in these mice remained low and did not differ from those detected in mice that did not receive the Clostridiales cocktail (Supplementary Fig. 11). Notably, a higher recovery of the administered Clostridiales was associated with a significantly lower susceptibility to MRKP gut colonization (Fig. 4d, two-sided Mann–Whitney test, $p = 0.026$). Altogether these results suggest that one of the mechanisms by which *Lactobacillus* restricts MRKP colonization is by promoting the expansion of Clostridiales taxa, which then diminish MRKP intestinal colonization.

**Metabolomic changes upon *Lactobacillus* administration generate a hostile environment for multidrug-resistant *K. pneumoniae* growth**

We next studied if *Lactobacillus* administration and the associated microbiota changes (i.e. Clostridiales expansion) would translate into changes in the gut environment that could impair MRE growth. To this end, we analyzed 2 weeks after antibiotic cessation, through nuclear magnetic resonance (NMR), the metabolome of antibiotic-treated mice that received *Lactobacillus* and compared it to that one of AVN-treated mice. Notably, the metabolite whose abundance differed most between *Lactobacillus*-treated mice and controls was butyrate (Fig. 5a, Supplementary Data File 10, two-sided Mann–Whitney and Benjamini–Hockberg correction, $p = 2.4e{-}5$, $q = 7.5e{-}4$). Increased butyrate was found following administration of both *L. murinus* and *L.*

*rhamnosus* (Supplementary Fig. 12). Interestingly, previous studies performed in mice support the notion that Clostridiales can diminish *Enterobacteriaceae* colonization through production of butyrate[25,26], a metabolite that can directly impair *Enterobacteriaceae* growth[27]. Consistent with a potential role of butyrate in MRKP inhibition, the levels of butyrate pre-MRKP inoculation negatively correlated with the capacity of MRKP to colonize the intestine (Fig. 5b; Two-sided Spearman test; $\rho = -0.731$; $p = 7.2e{-}9$).

In addition to the increase in butyrate levels, changes in another 6 metabolites were detected both in mice that received *L. murinus* and *L. rhamnosus* (Supplementary Fig. 13, two-sided Mann–Whitney test and and Benjamini–Hockberg correction, $p < 0.05$, $q < 0.1$, the exact p and q values are shown in Supplementary Data File 10). Interestingly, we detected a decrease in specific nutrients (i.e. glucose, threonine and serine, Fig. 5a) that previous studies have shown to support the growth of other *Enterobacteriaceae* strains in mice[28,29]. Consistent with a potential role of these nutrients in promoting in vivo growth of the MRKP strain used in this study, the levels of glucose, threonine and serine pre-MRKP inoculation positively correlated with the subsequent capacity of MRKP to colonize the gut (Fig. 5b).

Altogether, these results suggest that *Lactobacillus* administration and the subsequent expansion of specific commensal bacteria, mainly from the Clostridiales order, generate a hostile gut environment for MRKP growth, including the production of potential inhibitory molecules (i.e. butyrate) and the reduction of potential nutrient sources (i.e. serine, threonine, and glucose). In support of this hypothesis, we measured MRKP growth in filtered anaerobic cecal cultures of AVN-treated mice that received *Lactobacillus* and saw that it was significantly impaired (Fig. 5c). In addition, in vitro MRKP growth at the pH detected in the murine large intestine (pH = 6.5, Supplementary Fig. 14) was inhibited by butyrate in a dose-dependent manner (Fig. 5d), while serine, threonine, and glucose promoted MRKP growth (Fig. 5e).

Next, we attempted to identify the commensal bacteria responsible for the observed metabolomic changes. To this end, we performed a correlation analysis between the relative abundance of

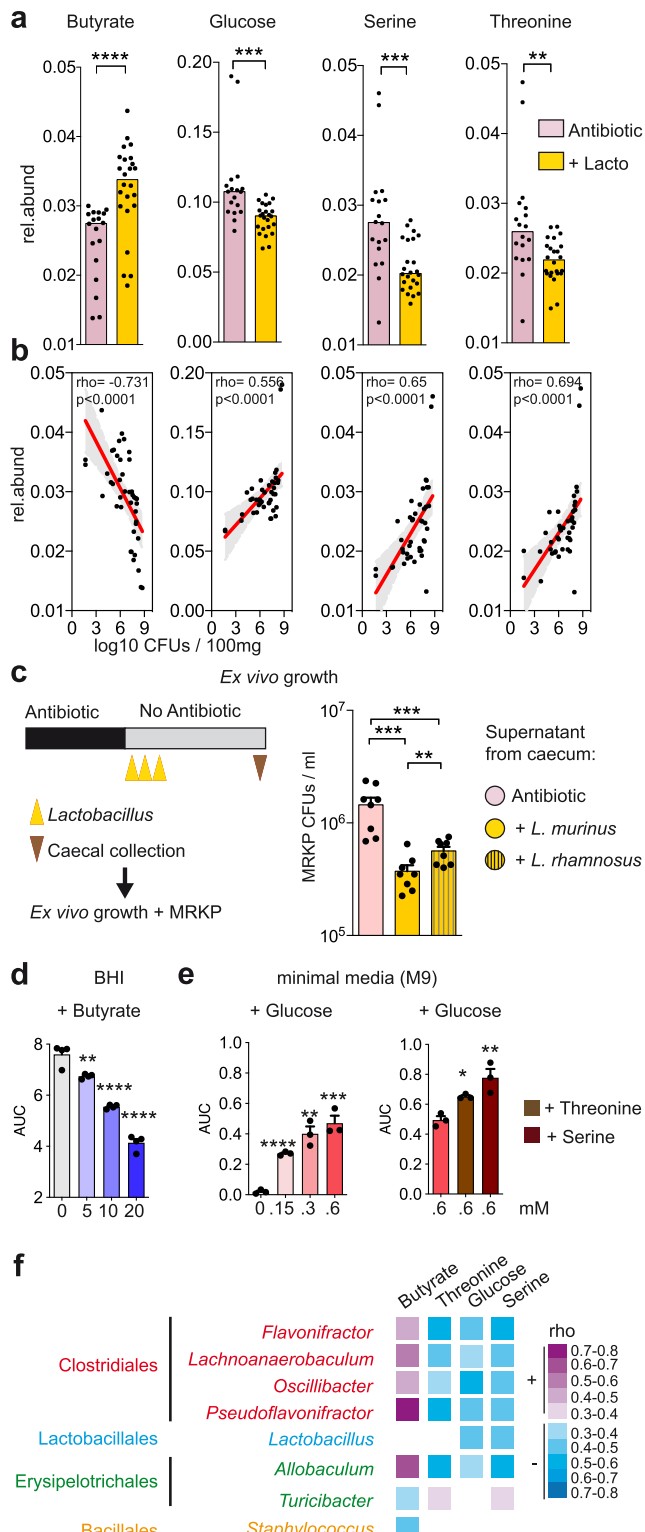

**Fig. 5 | Metabolomic changes upon *Lactobacillus* administration generate a hostile environment for multidrug-resistant *K. pneumoniae* (MRKP) growth. a** The levels of butyrate increased significantly, while the levels of serine, glucose and threonine decreased after administration of *Lactobacillus* strains to AVN-treated animals. Two-sided Mann–Whitney test, ****$p = 2.4e{-}5$,***$p < 0.001$,**$p = 0.002$ and Benjamini–Hockberg correction ($q < 0.01$), $N = 18$ and 24 mice per group. Rel.abund: relative abundance. Lacto: *Lactobacillus*. **b** Correlation analysis of the butyrate, serine, glucose and threonine levels pre-MRKP inoculation with the fecal MRKP levels one day after inoculation. Two-sided Spearman correlation test, ****$p < 0.0001$, $N = 42$ mice. **c** Cecal contents from AVN-treated mice that received *Lactobacillus* strains or bacterial vehicle were collected 2 weeks after antibiotic cessation and grown for 24 h. Subsequently MRKP was grown for 24 h in the filtered supernatant of the cecal cultures and its levels were quantified. Two-sided Mann–Whitney test, ***$p < 0.001$,**$p = 0.009$, $N = 4$ cecal contents per group and 2 replicates per cecal content. CFUs colony-forming units. **d** Butyrate inhibited MRKP growth in vitro in a dose-dependent manner. AUC area under the bacterial growth curve. Two-sided *T*-test compared to the control, ****$p < 0.0001$, **$p = 0.006$, $N = 4$ bacterial cultures per butyrate concentration tested. **e** MRKP can grow in minimal media containing glucose as the unique carbon source, while serine and threonine boost this growth. Two-sided *T*-test compared to the control, ****$p < 0.01$, ***$p = 0.0008$, **$p < 0.01$, *$p = 0.011$, $N = 3$ bacterial cultures per glucose, serine and threonine concentration tested. **f** Correlation analysis between the relative abundance of most prevalent genera (median > 0.01%) and fecal levels of butyrate, threonine, glucose or serine. Only genera with a significant association are shown. Two-sided Spearman correlation test and Benjamini–Hockberg correction, $p < 0.05$, $q < 0.1$, $N = 42$ mice. Only bacteria that could be classified up to the genus level are shown. All correlations are shown in Supplementary Data File 11. The correlation coefficient rho is color-coded, where the purple and blue colors represent positive and negative correlations, respectively. Missing boxes represent non-significant correlations. Bars represent the median in **a** and the mean in **c**–**e**. Whiskers indicate the SEM. The line in B represents the linear regression mean and the gray shadow the 95% CI. Source data are provided as a Source Data file.

*Oscillibacter* and *Pseudoflavonifractor*) and the genus *Allobaculum*, whose recovery was also higher after *Lactobacillus* administration (Fig. 5f, Two-sided Spearman correlation test and Benjamini–Hockberg correction; $p < 0.01$, $q < 0.01$, the exact $p$ and $q$ values are shown in Supplementary Data File 11). These data suggest that the commensals recovered from the residual microbiota of mice receiving *Lactobacillus* were responsible for the higher butyrate levels detected. In contrast, serine, threonine and glucose levels were negatively associated with the relative abundance of commensals that expanded upon *Lactobacillus* administration (Fig. 5f), suggesting that the recovery of these bacteria diminished the levels of nutrients available for MRKP growth. Nevertheless, a negative association was also detected between *Lactobacillus*, glucose and serine, indicating that *Lactobacillus* may directly contribute to reduce the levels of these nutrient sources.

Altogether, these results support a cooperative mechanism by which multiple commensals restrict the intestinal colonization by a multidrug-resistant pathogen. *Lactobacillus* promote changes in the microbiota, mostly the expansion of Clostridiales genera, that lead to changes in specific metabolites which negatively impact MRKP growth.

## Similar interactions between *Lactobacillus*, Clostridiales genera, butyrate and MRE levels are detected in acute leukemia hospitalized patients

Based on the results obtained in mice, we analyzed the fecal metabolome of our patients' cohort to study if similar mechanisms of protection against MRE intestinal colonization were relevant in humans. Specifically, we correlated the levels of butyrate, serine, threonine, and glucose with the levels of MRE in colonized patients. We could not analyze serine in patients, since none of the NMR peaks could be attributed exclusively to serine. Although a significant association was not detected between threonine or glucose and MRE levels, a significant negative correlation was found between the levels of butyrate and the levels of MRE (Fig. 6a, b; two-sided Spearman correlation test,

commensal bacteria and the levels of metabolites. There was no significant correlation between the levels of butyrate and the levels of *Lactobacillus* (Fig. 5f, Two-sided Spearman correlation test, $p = 0.286$). This lack of correlation was perhaps expected since butyrate is not considered a major product of *Lactobacillus* metabolism and since the two *Lactobacillus* strains that we used in our experiments did not produce butyrate when tested in vitro (Supplementary Fig. 15). In contrast, we saw significant positive correlations between the levels of butyrate and the Clostridiales genera that increased upon *Lactobacillus* administration (i.e. *Flavonifractor*, *Lachnoanaerobaculum*,

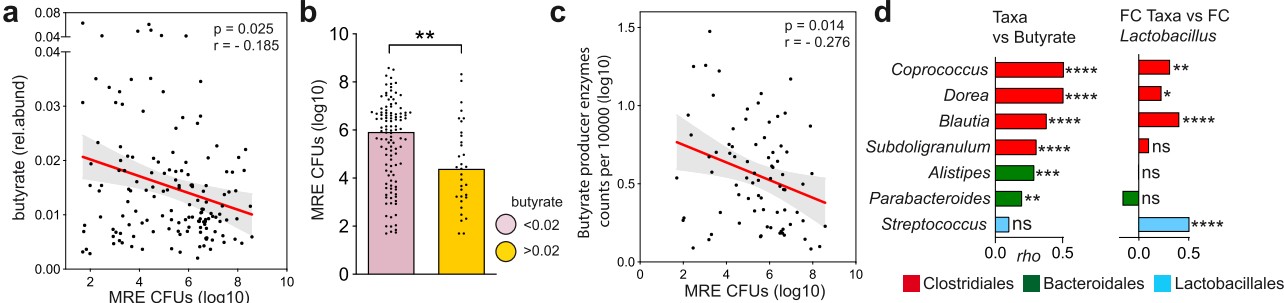

**Fig. 6 | Similar interactions between *Lactobacillus*, Clostridiales genera, butyrate and multidrug-resistant *Enterobacteriaceae* (MRE) levels are detected in hospitalized acute leukemia patients. a** Negative correlation between butyrate and MRE levels was detected in MRE-colonized samples collected from hospitalized leukemia patients. Two-sided Spearman correlation test, $p = 0.025$, $\rho = -0.185$, $N = 147$ samples. Rel.abund: relative abundance. CFUs colony-forming units. **b** MRE-colonized samples collected from hospitalized patients that had higher levels of butyrate had significantly lower MRE levels, as compared to samples that had lower butyrate levels. Two-sided Mann–Whitney test, \*\*$p = 0.005$; $N = 114$ and 33 samples per group. Bars represent the median. **c** Spearman correlation analysis of the fecal levels of enzymes involved in production of butyrate and the fecal levels of MRE in MRE-colonized samples. Only samples containing detectable levels of enzymes involved in production of butyrate (see methods) were included in the analysis. Two-sided Spearman correlation test, $p = 0.014$, $N = 78$ samples. **d** Correlation between (i) the abundance of the most abundant genera (median > 0.01%) and the levels of butyrate ($N = 175$ samples) or (ii) the changes in the abundance of most abundant genera and the changes in the abundance of the genus *Lactobacillus* ($N = 94$ pairs of samples). Only genera with a significant association are shown. Two-sided Spearman correlation test, \*\*\*\*$p < 0.0001$, \*\*\*$p = 0.0001$, \*\*$p < 0.01$, \*$p = 0.026$, and Benjamini–Hockberg correction ($q < 0.1$). Only bacteria that could be classified up to the genus level are shown. All correlations are shown in Supplementary Data File 12. Bars represent the rho values. The line in **a** and **c** represents the linear regression mean and the gray shadow the 95% CI. Source data are provided as a Source Data file.

$p = 0.025$; two-sided Mann–Whitney test, $p = 0.005$). Since other SCFAs have been shown to impact *Enterobacteriaceae* growth[12,30], we also investigated the potential impact of acetate and propionate on MRE levels. However, no significant correlation—positive or negative— was detected between these SCFA and MRE levels (Supplementary Fig. 16). These results suggest a negative and specific effect of butyrate on MRE intestinal colonization in patients.

In order to support the role of butyrate in protection against MRE intestinal colonization in patients, we made use of previously obtained proteomic data from the analyzed fecal samples[31]. We identified the peptides matching the enzymes required in the last step of butyrate synthesis (i.e. KEGG orthologs: K00929: *buk*, K01034: *ato*D, and K01035: *ato*A) and quantified their abundance in the fecal samples from ALP. As expected, the levels of these enzymes correlated positively with butyrate levels (Supplementary Fig. 17, two-sided Spearman correlation test, $\rho = 0.245$, $p = 0.02$). Most importantly, consistent with an inhibitory effect of butyrate on MRE colonization, the fecal levels of these enzymes were negatively associated with the fecal levels of MRE (Fig. 6c, Two-sided Spearman correlation test, $\rho = -0.276$, $p = 0.014$).

We next attempted to identify the potential bacteria responsible for the production of butyrate in our patients' cohort. To this end, we performed correlation analysis between the levels of butyrate and the levels of abundant genera (median > 0.01%) found in human samples. In concordance with previous studies[32], the commensal bacteria with the highest correlation with butyrate levels were from the order Clostridiales, the three top genera being *Coprococcus*, *Dorea*, and *Blautia* (Fig. 6d, two-sided Spearman correlation test and Benjamini–Hockberg correction, $p < 1e{-}6$, $q < 1e{-}6$, the exact $p$ and $q$ values are shown in Supp. Data File 12). Finally, we studied if *Lactobacillus* could be promoting the expansion of Clostridiales genera, which, as we have shown in mice, could directly restrict MRE intestinal colonization in part by increasing the levels of butyrate. To this end, we compared the dynamics of *Lactobacillus* genus with the dynamics of abundant genera found in ALP and found significant positive associations between the log2FC of *Lactobacillus* abundance and the log2FC of the abundance of the three Clostridiales genera which showed the highest positive association with butyrate levels (i.e. *Blautia, Coprococcus, Dorea*) (Fig. 6d, two-sided Spearman correlation test and Benjamini–Hockberg correction, $p < 0.05$, $q < 0.1$, the exact $p$ and $q$

values are shown in Supp. Data File 12). Consistent with a potential protective effect of the expansion of these three genera on MRE intestinal levels, the sum of the abundances of these three genera negatively correlated with MRE levels in colonized samples (Two-sided Spearman correlation test, $\rho = -0.26$; $p = 0.001$).

Altogether, our results suggest that *Lactobacillus* promote the expansion of specific Clostridiales commensals, both in mice and patients, which then impair intestinal colonization by MRE partially through production of butyrate.

## Discussion

Here we identified a mechanism by which the microbiome restricts gut colonization by MRE that involves cooperation between two common gut commensals: *Lactobacillus* and Clostridiales. Our results indicate that *Lactobacillus* promote the expansion of Clostridiales commensals after antibiotic treatment, which results in an adverse environment for MRE growth, characterized by higher levels of butyrate, a metabolite that can directly inhibit MRKP growth. This mechanism—an interaction between *Lactobacillus sp.* and Clostridiales is conserved in patients and mice.

Most mechanisms of colonization resistance against pathogens found so far involve either a single bacterium or groups of related bacteria. For example, *Klebsiella michiganensis*, a murine commensal bacterium, is a single organism that suffices to confer protection against pathogenic *Salmonella* Typhimurium in mice through nutrient competition[33]. However, the intestinal microbiota is a complex community of functionally and phylogenetically distant bacteria that interact and depend on each other[34]. It is plausible that many mechanisms of colonization resistance might require interaction and that interaction may involve microbes of phylogenetically distant taxa. We have identified one such cooperative mechanism where *Lactobacillus* enhance expansion of Clostridiales genera, which restricts MRE gut colonization. Similar to our results, a recent study demonstrated that cooperation between unrelated commensal bacteria (i.e. *Parabacteroides distasonis*, *Bacteroides sartorii*, *Blautia producta* and *Clostridium bolteae*) was required for restoring colonization resistance against VRE in mice[35]. That study showed that *B. producta* directly inhibits VRE growth, while the other bacteria enable intestinal colonization of *B. producta*. The cooperation, however, was not validated in patients.

*Lactobacillus* strains have been widely used as probiotics in order to restrict gut colonization by multiple pathogens[36]. Previous studies attempted to prevent gut colonization by antibiotic-resistant *Enterobacteriaceae* through the administration of *Lactobacillus* strains[37–39]. However, the results varied between studies, which raised questions regarding the protective effect of *Lactobacillus*. For example, administration of *Lactobacillus* strains provided colonization resistance against multidrug-resistant *E. coli* in mice[38]. In contrast, administration of a probiotic cocktail that included several *Lactobacillus* strains did not enhance the clearance of antibiotic-resistant *Enterobacteriaceae* in patients[39]. Thus, from these studies, it remained unclear the extent to which commonly used probiotics, such as *Lactobacillus*, can be useful in conferring colonization resistance against antibiotic-resistant *Enterobacteriaceae*. Our results provide a likely explanation for this problem: the protective effect of certain *Lactobacillus* strains depends on other commensals (i.e. Clostridiales bacteria). Our results are in agreement with previous work done in GFM in which administration of both *Lactobacillus* and spore-forming bacteria (which include among others, genera from the order Clostridiales) was required for inhibition of *E. coli* intestinal colonization[40]. Thus, future trials testing the effect of probiotics against infections should consider the interaction among different members of the microbiome and inter-subject microbiome variability.

Clostridiales have been shown to be a key bacterial group for restricting gut colonization by *Enterobacteriaceae* pathogens in mice[25,41]. However, the effect of Clostridiales species on MRE clinical isolates and, most importantly, in humans, remained unclear. Here, using a mouse model, we were able to confirm that Clostridiales have a negative impact on gut colonization by a clinical *Enterobacteriaceae* strain. Moreover, we analyzed patient data and found that specific Clostridiales taxa, whose expansion was associated with *Lactobacillus* increase, negatively correlate with MRE levels. This result suggests that the negative interaction between Clostridiales and *Enterobacteriaceae* is also conserved in patients. Clostridiales inhibition of pathogens depends on the production of butyrate[25,26], a major product of Clostridiales fermentation. Butyrate can inhibit the expansion of *Enterobacteriaceae* by inducing the PPARγ receptor in colonocytes, which limits the availability of oxygen and nitrate (two molecules that facilitate the intestinal growth of *Enterobacteriaceae*)[25,26]. In addition, butyrate can reduce the intracellular pH of bacteria, which can directly impair the growth of resistant *Enterobacteriaceae*[12]. Here, we have confirmed that butyrate can directly inhibit the growth of the MRE isolate used in our study. In addition, we detected a negative association between the levels of butyrate and butyrate-producing enzymes with MRE levels in hospitalized patients, a result that has not been previously reported. Altogether, these results suggest that butyrate produced by Clostridiales could contribute to MRE inhibition in patients as well as in mice.

Although butyrate was the most significant change detected in mice upon *Lactobacillus* administration, we also detected a significant change in other metabolites (Supplementary Fig. 13). The levels of serine, threonine and glucose decreased upon *Lactobacillus* administration. Serine, threonine and glucose are some of the preferred nutrients consumed by *Enterobacteriaceae* in the murine gut and both serine and glucose have been shown to promote *Enterobacteriaceae* growth in the murine intestine[28,29]. Here, we have confirmed that glucose, serine and threonine boost MRKP growth in vitro. Moreover, correlation analysis suggests that their reduction upon *Lactobacillus* administration could contribute to restricting MRKP gut colonization. Correlation analysis further indicates that commensal bacteria that recover upon *Lactobacillus* administration (mainly Clostridiales genera) could potentially be involved in the reduction of these nutrient sources. Consistent with this, a previous study performed in mice showed that administration of a Clostridiales strain from the genus *Blautia* can diminish the intestinal levels of threonine and serine[28].

Nevertheless, future studies should elucidate the contribution of different microbial taxa (including Clostridiales and *Lactobacillus*) on the reduction of these nutrient sources and to measure the extent these contribute to MRKP growth.

Our study suggests that administration of *Lactobacillus* could be an approach for restoring a major component of the microbiome (Clostridiales), not only in mice but also in hospitalized patients. This is consistent with previous data in which administration of a *L. rhamnosus*-supplemented formula resulted in expansion of several butyrate-producing strains in allergic infants, including some Clostridiales such as *Blautia, Roseburia,* and *Coprococcus*[42]. In the context of acute leukemia patients, expansion of Clostridiales by the administration of *Lactobacillus* could have a double positive effect: (i) restriction of MRE, (ii) induction of Tregs which may be beneficial for inflammatory disorders such as graft-versus-host disease that frequently occurs in these patients[43]. In this sense, identifying the specific mechanism by which *Lactobacillus* promote the recovery of Clostridiales could benefit the design of novel approaches that facilitate Clostridiales expansion. One potential mechanism would involve cross-feeding (the products derived from the metabolism of one bacterium are used for growth by another bacterium). Consistent with this mechanism, we detected an increase in specific molecules when *Lactobacillus* was cultured in vitro (ribose, adenine, lactate) and which boosted the growth of *Flavonifractor* in vitro, one of the Clostridiales that expanded in vivo after *Lactobacillus* administration. Besides facilitating Clostridiales expansion, molecules secreted by *Lactobacillus* could also alter Clostridiales metabolism. Interestingly, one of the *Lactobacillus*-produced metabolites that we detected in vitro (lactate) has been shown to be used by some Clostridiales genera for producing butyrate[44]. Thus *Lactobacillus* could be (1) promoting Clostridiales recovery but also (2) stimulating the butyrate production through secretion of lactate, which ultimately may enhance the capacity of Clostridiales to inhibit MRE. In line with this last hypothesis, when Clostridiales strains were administered to antibiotic-treated mice, MRKP colonization was significantly restricted, but slightly less than what we observed when *Lactobacillus* was given to mice (Fig. 3c vs Fig. 4d). However, it is also possible that the Clostridiales strains that we administered did not have the same inhibitory capacity as endogenous Clostridiales that expanded after *Lactobacillus* administration. Future work, including in vivo studies in mice and humans and the generation of *Lactobacillus* mutants defective in specific metabolic pathways should be performed to confirm the role of lactate and other metabolites in facilitating colonization resistance against antibiotic-resistant pathogens through interactions between *Lactobacillus* and different Clostridiales strains. In addition, future studies should evaluate if the interaction of *Lactobacillus* and Clostridiales is involved in protection against other pathogens besides MRE. In this sense, a recently published study identified that the administration of a probiotic cocktail containing 3 *Lactobacillus* strains diminished the rate of *C. difficile* infection in hospitalized patients[45]. It will be interesting to evaluate if in this case *Lactobacillus* can directly inhibit *C. difficile* or the expansion of Clostridiales species is responsible for this beneficial effect. For example, promoting the expansion of *Clostridium scindens*, a Clostridiales commensal that can inhibit *C. difficile* through secondary bile acid production[22].

We note some limitations to our study: (i) We have shown that *Lactobacillus* and Clostridiales can diminish gut colonization by MRKP. However, these do not provide complete restoration of colonization resistance, suggesting some other mechanisms might be at play. Nevertheless, several studies suggest that reduction and not complete elimination of MRE levels can significantly decrease the risk of bloodstream dissemination[6,46,47]. Thus the identification, as we have done in this study, of human commensal bacteria capable of reducing MRE intestinal levels could be useful for the development of alternative approaches to prevent infections by these type of pathogens.

(ii) 16S rRNA sequencing does not have enough resolution to characterize commensal bacteria at the strain level which may have precluded the discovery of additional mechanisms of protection. Nevertheless, 16S rRNA analyses have allowed us to identify interactions (validated in our mouse model) between bacterial taxa that are very prevalent in the human population. Thus the cooperative mechanism of resistance found in this study is likely present in a higher number of hospitalized patients, as opposed to mechanisms that are strain-specific. (iii) We have shown that *Lactobacillus* can have a beneficial effect for hospitalized patients treated with multiple antibiotics and predisposed to infections with multidrug-resistant pathogens. However, in healthier individuals, administration of a probiotic cocktail including *Lactobacillus* has been shown to impair microbiota compositional recovery[48]. Thus, the usage of *Lactobacillus* should be restricted to cases of major dysbiosis in which the recovery of specific taxa (i.e. Clostridiales) is impaired and can lead to life-threatening diseases.

In summary, our work describes a mechanism by which the interplay between two distant members of the microbiome restricts intestinal colonization by an antibiotic-resistant bacterial pathogen. This is the first study that describes this cooperative mechanism both in mice and in patients. Our results highlight the relevance of studying the interactions between different members of the microbiota in order to design efficient probiotic therapies.

## Methods

All mouse procedures including specific-pathogen-free mice were performed in accordance with institutional protocol guidelines at the "Servei Central de Suport a la Investigació Experimental" at the University of Valencia. Mice were maintained accordingly to the National guidelines (RD 53/2013), under protocols approved by University of Valencia Animal Care Committee describing experiments specific for this study. All mice procedures with GFM were performed at Instituto Gulbenkian de Ciência (IGC) and were approved by the Institutional Ethics Committee and the Portuguese National Entity (Direção Geral de Alimentação e Veterinária; Ref. number 015190), which complies with European Directive 86/609/EEC of the European Council. The study involving patients was conducted in accordance with the Declaration of Helsinki, and the protocol was approved by the Ethics Committee of CEIC Dirección General de Salud Pública y Centro Superior de Investigación en Salud Pública (20130515/08). The study approved by the Ethics Committee was specifically designed to evaluate the role of the microbiome on the intestinal colonization by MRE. All included patients gave their consent to participate in the study. Participants did not receive any compensation for participating in the study.

### Animal experiments

All experiments with SPF mice were done with 7-week-old C57BL/6J female mice purchased from Charles River laboratories and housed with autoclave-sterilized food (a 1:1 mixture of 2014S Teklad Global diet and 2019S Teklad Global Extrused 19% Protein Rodent Diet from Envigo) and autoclave-sterilized water. Temperature was kept at $21 \pm 2\,°C$ and humidity was maintained at 60–70%, in 12 h light/dark cycles.

All gnotobiotic experiments were done with 7-week-old C57BL/6J female mice and were performed under aseptic and sterile conditions. Animals were maintained singly housed with food and water *ad libitum* inside airtight HEPA-filtered isocages placed in an isocage rack. The temperature was kept at $22\,°C \pm 2\,°C$ and humidity was maintained at $45–50\% \pm 10\%$, in 12 h light/dark cycles. Animal manipulation was performed in an ISOcage Biosafety Cabinet, under sterile conditions. Mice were fed with a rat and mouse number 3 breeding autoclaved diet from Special Diet Services.

### Animal experiments: effect of persistent changes in the microbiota on MRKP gut colonization (Fig. 2)

Mice received ampicillin (0.5 g/L), vancomycin (0.5 g/L) and neomycin (1 g/L) in the drinking water for 7 consecutive days. Subsequently, antibiotic treatment was stopped and mice were allowed to recover for 2 or 4 weeks before MRKP inoculation. Another group of mice received regular water instead. A maximum of two mice were housed in the same cage to minimize the number of mice from the same group sharing the cage, since their microbiomes would be very similar due to coprophagy. Fecal samples for microbiota analysis were collected immediately before MRKP inoculation. These fecal samples were stored at −80 °C until they were processed. Fecal samples for quantifying MRKP levels were collected immediately before MRKP inoculation and 24 h after inoculation. To this end, samples were diluted in PBS and dilutions were plated in Luria-Bertani (LB) agar supplemented with ampicillin (100 µg/ml), vancomycin (100 µg/ml) and neomycin (50 µg/ml). Plates were incubated at 37 °C during 24 h. Note that no colony-forming units (CFUs) were detected in the sample collected before MRKP inoculation neither in this nor in subsequent experiments.

### Animal experiments: effect of commensal bacteria administration on MRKP gut colonization (Figs. 3 and 4; Supplementary Figs. 6 and 7)

Mice received ampicillin (0.5 g/L), vancomycin (0.5 g/L) and neomycin (1 g/L) in the drinking water for 7 consecutive days. Subsequently, antibiotic treatment was stopped and mice were allowed to recover for 2 weeks before MRKP inoculation. One group of mice received by oral gavage 200 µl containing ~$10^6$ CFUs of specific commensal bacteria for 3 consecutive days, starting the day after antibiotic cessation. As specified in the results section these bacteria were *L. rhamnosus*, *L. murinus*, *Barnesiella* or a cocktail of Clostridiales isolates (*Oscillibacter*, *Flavonifractor*, Unclassified_*Ruminococcaceae* and *Blautia*). See below the methodology for the isolation, growth of the different isolates and preparation of the inoculum. As a control, another group of mice received PBS with 15% glycerol and 0.1% cysteine (PBS-GC, the vehicle for resuspending the bacteria) instead of the bacteria. Two weeks after antibiotic cessation, MRKP was inoculated through oral gavage as described above. Fecal samples for microbiota analysis were collected immediately before MRKP inoculation and for MRKP quantification after MRKP inoculation as described above. In addition, fecal samples were collected for metabolome analysis immediately before MRKP inoculation. Samples for microbiota and metabolome analysis were stored at −80 °C until samples were processed. In the experiment shown in Supplementary Fig. 6, samples from the colonic mucosa were also collected for MRKP quantification. Two days after MRKP inoculation, mice were euthanized. Three cetimeters of the colon were excised, and the contents were eliminated through manual extrusion. Subsequently, the colon was flushed twice with PBS and open longitudinally, the colonic mucosa was scrapped with a scalpel and resuspended in PBS. Dilutions were plated in selective media.

### Animal experiments: effect of *Lactobacillus* administration on MRKP gut colonization in GFM (Fig. 3d–f)

C57BL/6J mice bred under germ-free (GF) conditions in the animal house facility at the Instituto Gulbenkian de Ciência were used for gnotobiotic experiments. Prior to each experiment, mice were kept in individually ventilated ISOcages and were always manipulated inside an ISOcage Biosafety Cabinet to avoid environmental microbial contamination. Mice were gavaged with *L. rhamnosus* strain, *L. murinus* strain, or PBS (control group) prior to gavage with MRKP strain 10 days later, with loads being assessed the following day by selective plating in Man-Rogosa-Sharp (MRS) Agar (*Lactobacillus* strains) and LB (MRKP strain).

## Commensal bacteria isolation from murine samples

Since all the bacteria that we wanted to isolate were anaerobic, in order to preserve their viability, manipulations were performed inside an anaerobic chamber (Whitley DG250 Anaerobic Workstation, Don Whitley Scientific Limited), supplied with a 10% $CO_2$, 10% $H_2$, and 80% $N_2$ compressed gas mixture. The material used for bacterial growth (solid or liquid media) was deoxygenated overnight inside the anaerobic chamber prior to its use.

In order to grow and isolate intestinal anaerobic commensal bacteria from the mice, we collected the cecal content of 7-week-old SPF-C57BL/6 mice purchased from Charles River laboratories. The cecal content was immediately resuspended in PBS-cysteine 0.1% (approximately the same volume of cecal content as buffer). Subsequently, the tube containing the resuspended cecum content was transported to the laboratory in an anaerobic jar. In the laboratory, working in the anaerobic chamber, 7 ml of PBS-GC was added to the suspension. The suspension was frozen at −80 °C in aliquots of 150 µl. One aliquot was thawed and plated under anaerobic conditions on the culture media Columbia Blood Agar (CBA) (dilution from $10^{-2}$ to $10^{-6}$). Plates were incubated in the anaerobic chamber for 6 days at 37 °C. Grown colonies were re-streaked in a new CBA plate. Taxonomic identification of each isolate was performed through a colony-PCR of the 16S rRNA gene and Sanger-sequencing of the PCR product. The PCR was performed using the universal primers F27 (ACGAAGCAT-CAGAGTTTGATCMTGGCTCAG) and R1492 (CGGTTACCTTGTTAC-GACTT). Each PCR reaction was done in a total volume of 25 µl, using Thermopol® Reaction Buffer 10× (2.5 µl), dNTPs 10 mM (0.625 µl), primer forward 10 µM (0.5 µl), primer reverse 10 mM (0.5 µl), Taq-polymerase (0.5 µl), 1 or 5 µl of bacteria resuspended in PBS as DNA source (according to the concentration of the bacterial suspension) and ultrapure water to adjust the volume to a total of 25 µl. The parameters of PCR reaction: initial denaturation (5 min, 94 °C); 35 cycles of denaturation (30 s, 94 °C); hybridization (30 s, 56 °C); elongation (30 s, 68 °C). After the 35 cycles, the reaction finalized with an elongation cycle (5 min, 72 °C). PCR products were purified using purification plates ExcelaPure™ 96-well Ultrafiltration plate (Edge-Bio) and were sequenced through capillary electrophoresis on the sequencing facility of the Servei Central de Suport a la Investigació Experimental of the University of Valencia. The quality of retrieved sequences in ab.1 format was manually checked using Trev program from package Staden 2.0 (available at http://staden.sourceforge.net/) and sequences were kept in fasta format. Phylogenetic classification of the sequences was done using Mothur v.1.36 as described below[49]. The isolated and taxonomically characterized bacteria were stocked at −80 °C in PBS-glycerol 20%. Using this procedure, the strains CU100 (*L. murinus*), CU971 (*Oscillibacter*) and CU972 (Ruminococcaceae_Unclassified) were isolated. The *Barnesiella* murine isolate was kindly provided by Eric Pamer[22].

## Commensal bacteria isolation from human samples

Fecal samples from hospitalized acute leukemia patients were already collected and stored and −80 °C in a previous published study[17]. Each sample was inoculated in blood culture bottles (bioMérieux) containing 5 ml of rumen media filtered at 0.2 µm and 5 ml of 5% defibrinated sheep blood, then incubated at 37 °C under two atmospheric conditions (aerobic and anaerobic). At day 1, 3, 7, and 15 of incubation, the inoculums were plated on Columbia agar with 5% sheep blood and incubated at 37 °C for 24 to 72 h in aerobic and anaerobic atmosphere. Based on the morphological aspect of the colonies, the bacteria were selected and transplanted onto Columbia agar with 5% sheep blood, incubated at 37 °C for 24 to 72 h in aerobic and anaerobic atmosphere and then purified bacteria were identified using matrix-assisted laser desorption ionization-time of flight mass spectrometry (MALDI-TOF MS) (Microflex, Bruker Daltonics, Bremen, Germany). The isolated and taxonomically characterized bacteria were stocked at −80 °C in PBS-

glycerol 20%. Subsequently, 16S rRNA PCR sequencing and comparison with the NCBI RefSeq database was applied to confirm the taxonomy of the isolated strains. By performing this procedure, the strains CU864 (*Flavonifractor plautii)* CU826 (*B. producta*) and CU700 (*L. rhamnosus*) were isolated.

## Commensal bacteria inoculum preparation

For the preparation of the bacterial inoculum, each bacterial strain was cultivated in the corresponding media in an anaerobic chamber at 37 °C. Strains belonging to *Lactobacillus* were grown on MRS agar, while the other strains were grown in Columbia Blood agar plates. All necessary material was left in the anaerobic chamber for 24 h prior to its use in order to ensure anaerobic conditions. Bacteria were collected from plates and resuspended in PBS-GC, until they reached the same absorbance (optic density-OD600 of 0.1) for all the strains. Equal volume was taken from each strain suspension in order to produce a mix of commensal bacteria. Once prepared, the inoculums were aliquoted and frozen at −80 °C. The day of inoculation, the inoculums were placed and transported on dry ice to the animal facility. Inoculums were thawed immediately before their oral inoculation in mice.

## MRKP inoculum preparation

MRKP strain MH258 was obtained at Memorial Sloan-Kettering Cancer Center where it was isolated from a blood sample of a hospitalized patient[50], and was kindly provided by Eric Pamer. MRKP was grown at 37 °C in LB agar overnight. The next morning it was resuspended in PBS. $10^6$ CFUs of MRKP were administered to each mouse by oral gavage in a 200 µl volume.

## Ex vivo experiment to study the growth capability of MRKP in supernatants of cultured cecal contents

Mice received ampicillin (0.5 g/L), vancomycin (0.5 g/L) and neomycin (1 g/L) for 7 consecutive days in drinking water. Subsequently, antibiotic treatment was stopped and mice were allowed to recover for 2 weeks. One group of mice received 200 µl containing -$10^6$ CFUs of *L. rhamnosus* or *L. murinus* by oral gavage for 3 consecutive days, starting the day after antibiotic cessation. As a control, another group of mice received PBS-GC instead of the bacteria. Two weeks after antibiotic cessation, the cecal content of mice was collected and introduced in anaerobic jars until it was introduced in an anaerobic chamber. Inside the anaerobic chamber, 250 mg of cecal content were resuspended in 1 ml of PBS (previously reduced in the anaerobic chamber) and incubated for 24 h at 37 °C, under anaerobic conditions. Subsequently the culture was centrifuged at 10,000 × *g* for 3 min. Next, the supernatant was filter-sterilized using first 0.45 µm pore PVDF microcentrifuge filters and then 0.2 µm pore filters. For each cecum, 35 µl of filtered supernatants were inoculated with 5 µl of MRKP PBS suspension containing $10^3$ CFUs. Two replicates were prepared for each condition and incubated in anaerobiosis for 24 h at 37 °C. After incubation, dilutions of the cultures in PBS were plated on LB with ampicillin (100 µg/ml) and neomycin (50 µg/ml) and incubated overnight to quantify MRKP CFUs.

## In vitro assay to study the growth capabilities of MRKP in the presence of butyrate, glucose, serine or threonine

To test the inhibitory properties of butyrate against MRKP growth, MRKP was grown in Brain Heart Infusion broth (BHI) at pH = 6.5 containing 20, 10, 5, or 0 mM of butyrate. The inoculum of MRKP was prepared by suspending MRKP in BHI at an absorbance of 0.1. Butyrate concentrations used in this assay are based on the butyrate levels that have been found in murine and human large intestinal samples[12,51].

To test the capacity of MRKP to grow on glucose as unique carbon source, MRKP was grown in minimal media M9 containing increasing concentrations of glucose (0.15 mM, 0.3 mM, 0.6 mM). Used concentrations were based on those found in the large intestine of mice[28].

The inoculum of MRKP was prepared by suspending MRKP in M9 at an absorbance of 0.1. To evaluate if MRKP growth could be enhanced by the presence of threonine or serine, MRKP was grown in minimal media M9 containing glucose 0.6 mM and the same concentration of serine or threonine.

All cultures were performed in 96-well plates containing a final volume of 100 µl. Cultures were grown at 37 °C on a Tecan infinite M plex during 20 h incubation. Optical density (OD) was measured every hour. The OD value from wells without bacterial inoculum was subtracted from those values obtained in wells containing bacteria. The area under the curve was calculated using Graphapad 6.0.

### In vitro assay to study the growth capabilities of a Clostridiales strain in the presence of metabolites produced by *Lactobacillus*

For testing the effect of metabolites found to be produced by *Lactobacillus* strains on the growth of Clostridiales, we used a strain from the genus *Flavonifractor*, which can be grown in BHI, the media in which the metabolites produced by *Lactobacillus* were detected. The *Flavonifractor* strain was grown in Columbia blood agar plates for 48 h under anaerobic conditions. Subsequently the colonies were resuspended in pre-reduced BHI inside of the anaerobic chamber at 37 °C. One hundred microliters of BHI containing 100 mM of DL-lactate, 1 g/l of adenine and 5 g/l of D-ribose was mixed with 100 µl of *Flavoniflactor*, making the final tested concentrations 50 mM DL-lactate, 0.5 g/l adenine and 2.5 g/l D-ribose, concentrations within the range of those previously used to test the effect of these metabolites on the growth of commensal microbes[44,52,53]. The strain was grown in triplicate. The growth curve was formed by measuring absorbance at OD620 every 20 min on the plate reader inside the anaerobic chamber and at 37 °C. As control, the *Flavonifractor* strain was also grown in BHI media without the addition of the specific metabolites.

### Fecal sample collection for analyzing how changes in the microbiome are associated with intestinal colonization by MRE in hospitalized patients

Fecal samples were obtained from a previous published study in which we analyzed how clinical factors influence MRE intestinal colonization in acute leukemia patients[17]. Samples were collected weekly from December of 2013 until May of 2015 from all acute leukemia patients hospitalized at the Hospital La Fe (Valencia, Spain) who agreed to participate in this study. Samples were kept at 4 °C for less than 24 h. Subsequently, three aliquots of each sample were weighted and resuspended in 1 ml of autoclave-sterilized PBS 15% glycerol in order to preserve viability of bacteria upon freezing and kept at −80 °C until further processing. Three other aliquots were not resuspended in PBS and directly frozen at −80 °C for subsequent proteomic, metabolomic or metagenomic analysis. Each aliquot contained approximately 200 mg of fecal material. MRE levels in each collected fecal sample (PBS-glycerol frozen material) were analyzed in the previous published study by culturing the fecal samples in selective media[17]. In addition, taxonomic identification of isolated MRE colonies was performed through MALDI-TOF MS. Moreover, the antibiotic resistance pattern was determined through the Vitek 2 system. In this particular study, we analyzed the microbiome and metabolome of a subset of samples. Specifically, within all the 133 acute leukemia patients for which fecal samples were collected in the previous study[17], we chose a subset of 58 patients for microbiota and metabolomic analysis since we wanted to specifically evaluate the dynamics of the microbiome and its association with changes in MRE levels. The criteria to select the patients were the following: we included patients in which MRE colonization was detected in at least one sample and for which we were able to collect at least a follow-up sample during the same hospital admission (in most cases this sample was collected 1 week after the first sample). Additional consecutive samples from that patient were also included for its microbiota analysis until MRE was not detectable or the patient was discharged. In the case of an MRE recolonization during the same or another hospitalization period, samples from this new colonization process were also included following the same criteria. A total of 225 samples match this criteria and were processed for microbiota analysis (16S rRNA). Two hundred sixteen of these samples for which we had enough material were also analyzed through Nuclear Magnetic Resonance as described below. In addition, 212 of the 225 samples for which we had enough material were analyzed by proteomics, as described in a previous publication[31].

In the previously published study[17], we analyzed the levels of MRE colonization in fecal samples from acute leukemia patients, characterized their taxonomy and resistance to antibiotics and investigate the associations between MRE levels and different clinical variables, including antibiotics and antifungals, neutropenia, mucositis and parenteral feeding[17]. We demonstrated that beta-lactam administration significantly reduced the fecal levels of MRE strains sensitive to the administered beta-lactam[17]. In order to avoid this confounding variable, we excluded pairs of samples collected during the period beta-lactams were given to the patient and detected MRE were sensitive to beta-lactams. This produced a total of 179 samples from 47 patients that were used in the analysis shown in this study. The characteristics of these patients (sex and age are shown in Supplementary Data file 13).

### Fecal DNA extraction

Bacterial DNA from fecal samples was extracted using the QIAamp® DNA Fast Stool Mini kit (QIAGEN, Spain). Extractions were performed according to manufacturer instructions with introduction of a previous mechanic disruption step with bead-beating. Briefly, samples (-200 mg for human samples and -50 mg for mouse samples) were resuspended in the first buffer of the extraction kit. Subsequently the samples were shaken on a Vortex-Genie 2 equipped with a Vortex Adapters (Mobio) at maximum speed for 5 min in the presence of 500 µl of glass micro-beads (acid-washed glass beads 150–212 µm, Sigma®). Then, we followed the protocol of QIAamp® DNA Fast Stool Mini kit.

### 16S rRNA gene sequencing and analysis

The V3-V4 region of the 16S rRNA gene was amplified (Kapa HiFi HotStart Ready Mix), indexed with Nextera® XT Index Kit (96 indexes, 384 samples) and sequenced as described in the manual for "16S Metagenomic Sequencing Library Preparation" of the MiSeq platform (Illumina) using Miseq Reagent Kit V3.

Sequences were analyzed as we have previously described[31]. Quality assessment of the obtained sequences was performed using printseq-lite v.0.20.4. Sequences were trimmed using the sliding-window technique, such that the minimum average quality score over a window of 20 bases never dropped below 30. Sequences were trimmed from the 3′-end until this criterion was met. Then, trimmed forward and reverse paired-end sequences were assembled using fastq-join v.1.1.2 from ea-tools suite[54], applying default parameters (maximum 8% of difference and minimum overlap of 6 bp). Assembled paired-end sequences larger than 400 bp were kept for subsequent analysis. Sequences were aligned to the 16S rRNA gene database using the SILVA reference alignment[55] as the template, and the Needleman–Wunsch algorithm with the default scoring options. Potentially chimeric sequences were removed using the uchime algorithm[56], implemented in Mothur. To minimize the effect of sequencing errors in overestimating microbial diversity[57], rare abundance sequences that differ in 1% from a high abundance sequence were merged to the high abundance sequence using the pre.cluster option in Mothur[49]. Since different numbers of sequences per sample could lead to a different diversity (i.e., more Operational Taxonomic Units-OTUs could be obtained in those samples with higher coverage), we rarefied all samples to the number of sequences obtained in the sample with the lowest number of sequences (i.e. 10,095 for samples

included in the analysis of Figs. 1 and 6, 9810 for samples included in the analysis of Fig. 2, 16,285 for samples included in the analysis of Fig. 4). Sequences were grouped into OTUs using the vsearch algorithm, with the abundance-based agc method, implemented in Mothur. Sequences with distance-based similarity of 97% or greater were assigned to the same OTU. Singletons (very rare OTUs identified in only one sample, containing only one count) were not included in the analysis.

Phylogenetic classification of sequences was performed for each sequence using the Bayesian classifier algorithm and the Mothur-formatted RDP training set v.10 with the bootstrap cutoff 60%[58]. Classification was assigned to the genus level when possible; otherwise, the closest level of classification to the genus level was given, preceded by "unclassified; UC".

Shannon index was obtained at the OTU level using the package vegan v2.5–3 and the R 3.4.0 software. To perform the PCoA analysis in Fig. 1, we first calculated the log2FC between the abundance of genera identified in a pair of consecutive samples collected from the same patient. These log2FC was calculated for each genus identified in all patients and for each pair of consecutive samples collected from every patient. Next, the Euclidean distances between every sample pair were calculated using the vegan package. Euclidian distance was utilized instead of the more widely used Bray–Curtis distance since the log2FC table contained negative values which can lead to misleading results when calculating the Bray–Curtis distance. The obtained distance matrix was used for the PCoA analysis that was performed with the R package labdsv 2.0. In Fig. 2 the PCoA was performed with the Bray–Curtis distances instead since in this particular case the distances were not calculated with log2FC values but rather with count data obtained from the 16S rRNA sequencing analysis.

In order to study the prevalence of the OTU20 (*L. rhamnosus*) in other cohorts worldwide, we compared the representative sequence of the OTU20 against 16S rRNA databases of fecal samples from healthy individuals. To this end, we selected extensive databases (>500 individuals) sequenced using the same technology used in this study (i.e. MiSeq Illumina) and covering at least a part of the 16S region sequenced in our work (i.e. V3 and/or V4 regions). We downloaded sequences from the databases from the Flemish Gut Flora project (*N* = 1054, Belgium)[59], MORINAGA (*N* = 642, Japan) and NIBIONH (*N* = 954, Japan)[60], Milieu Intérieur project (*N* = 1311, France)[61] and the Spanish Gut Microbiome Project (*N* = 530, Spain)[62]. Quality assessment and assembly of pair-end sequences was performed as described above, which led a total of $2.4 \times 10^8$ sequences, average of 53707 per sample. Oligonucleotides used for 16S RNA amplification were removed using printseq-lite. Subsequently, the representative sequence of the OTU20 (the most abundant sequence within this OTU) was aligned against each sequence from each sample using blastN v.2.12. A sequence was assigned to the OTU20 when its similarity with the representative sequence of the OTU20 was >97% (standard threshold used for the definition of OTU) with a 100% coverage.

**Quantification of the microbiota bacterial density through qPCR of the 16S rRNA gene**

As a proxy of the total microbiota density, we quantified the number of 16S rRNA gene copies per sample by using universal primers previously recommended for their high coverage rates in all bacteria[63]: 16S U515F 5′-GTGCCAGCMGCCGCGGTAA-3′ and 16SU789 5′-TAGATACCCSSGTA GTCC-3′. Amplification was performed in a 20 µl final volume containing 0.5 µl of template DNA, 5 µl of the LightCycler 480 SYBR Green I Master Mix, 0.2 mM of each primer and 4.6 µl of nuclease-free water. The thermocycling protocol used was as follows: an initial step of 95 °C for 10 min and 35 cycles of 10 s at 95 °C, 20 s at 58 °C, and 20 s at 72 °C. Duplicates were performed for each sample.

The total bacteria levels in each sample were calculated by comparison with the Cq values obtained from a standard curve, applying

the same thermocycling protocol. The standard curve was generated in a similar manner as a previously published protocol[64], using serial ten-fold dilutions of DNA extracted from $10^8$ bacteria, pooled from 10 different species (*E. coli, K. pneumoniae, Listeria innocua, Prevotella intermedia, Streptococcus mutans, Streptococcus dentisani, Fusobacterium nucleatum, Pseudomonas aeruginosa, Actinomyces naeslundii* and *Veionella parvula*), which were quantified and sorted in a FACsaria III cytometer after mild sonication to separate aggregated cells.

**Genome sequencing and analysis**

The two isolated *Lactobacillus* strains were cultured in anaerobic conditions on MRS plates. Bacteria were resuspended in PBS and DNA was extracted using QIAamp® DNA Fast Stool Mini kit as described above. Subsequently, a genomic DNA library was obtained with the kit Nextera® XT DNA Sample Preparation Kit according to the manufacturer's guide. The samples were indexed with Nextera® XT Index Kit and the genome sequencing was performed with an Illumina MiSeq® System, using Miseq Reagent Kit V3 (pair-end).

Adapter sequences were removed from the raw data using Cutadapt v.1.10[65]. Sequences were then filtered by quality using UrQt v.1.0.18 (last update September 2016)[66]. Cleaned genomic data was assembled using SPAdes v.3.7.1 using the "careful" algorithm to improve the contig reconstruction[67]. A multi-kmer Bruijn graph reconstruction was then used as it has been suggested to improve the assembly outcome. We used 6 different kmer lengths (21, 33, 55, 77, 99, 127), as it is the best kmer combination for estimated read sizes over 250 bp. Open reading frames (ORFs) were identified and annotated using PROKKA v.1.13[68].

The taxonomic classification of the isolated bacteria was performed by calculating the core-genome-based average nucleotide identity against the NCBI RefSeq database (last accessed March 2018). On one side, the fasta sequence of 11175 complete bacterial genomes was downloaded from the RefSeq repository. Genomes were reannotated using PROKKA and grouped at species level according to the taxonomic assignation in source. We constructed the core genome of each species group with at least 4 genomes, following the methodologies previously described[69]. In brief, gene orthologs were identified as bidirectional best hits between genes from independent genomes, using end-gap free global alignment between the proteome of Lactobacilli species. Hits with less than 40% (genus) or 80% (species) similarity in amino acid sequence or more than 20% difference in protein length were discarded. The core genome was defined as the intersection of pairwise lists of strict positional orthologs present in all assessed genomes. Finally, a concatenate of core genes was performed for each genome.

On the other side, the genome assemblies of the isolated bacteria were split in fragments of 500 bp, overlapping in 150 bp. Fragments were then mapped against the core-genome concatenates, using bowtie2 v.2.2.9[70]. Only fragments with a coverage greater than 80% were considered. The core-genome Average Nucleotide Identity (ANI) was calculated by dividing the percentage of identity of each fragment by the total number of fragments mapping the core-genome concatenate and the length of the concatenate.

To reconstruct the *Lactobacillus* phylogeny, 115 genomes belonging to the genus *Lactobacillus* were collected from the RefSeq dataset and their core genome was reconstructed, including the two query genomes, following the methodologies previously described[69]. A core genome concatenate was built using the protein sequences and aligned using mafft v.7407 with the −linsi algorithm. The resulting multiple sequence alignment was used to construct a phylogenetic tree using iqtree2 v.2.1.0 with the model LG + G + I4 (selected as the best model for the dataset according to the Akaike Information criterion) and 1000 rapid bootstrap iterations. In total, 197 core genes were used at protein level to reconstruct the phylogeny. The root of

the tree was inferred using the genome from *Clostridium cuniculi*, a bacterial species phylogenetically distant from *Lactobacillus*, for which we had previously reported its sequence[71]. Final visualization was performed using FigTree v.1.4.3.

A similar analysis as described above (calculation of ANI score) was performed comparing the *L. rhamnosus* genome against the 198 *L. rhamnosus* genomes available at the NCBI RefSeq database (last accessed Dec 2021). In addition, a phylogeny of *L. rhamnosus* was reconstruct using the 198 *L. rhamnosus* genomes and the *L. rhamnosus* genome from this study. In this particular case, 236 core genes were used to reconstruct the phylogeny.

## Metabolomic analysis

Human fecal samples (20–50 mg) were homogenized using the FastPrep-24 tissuelyser (MP Biomedicals, Irvine, CA, USA) in 0.5 ml of phosphate buffer (0.2 M, pH 7.0) prepared in deuterium oxide and sodium 3-(trimethylsilyl)−2,2′,3,3′-tetradeuteropropionate (TSP, 1 mM). After centrifugation (10,000 × $g$, 10 min, 4 °C), supernatants were collected and the remaining pellet was further extracted once as described above. Supernatants obtained from two runs of extraction were combined and centrifuged at 10,000 × $g$ for 10 min at 4 °C. A total of 600 µl of supernatant was transferred into NMR tubes with an outer diameter of 5 mm.

Mice fecal pellets were resuspended in water and mixed with 200 µl of phosphate buffer (0.2 M, pH 7.0) prepared in deuterium oxide and containing TSP (1 mM). After centrifugation (5500 × $g$, 15 min, 4 °C), supernatants (600 µl) were transferred into NMR tubes.

$^1$H NMR spectra of fecal samples were obtained at 300 K using a Bruker Avance III HD 600 MHz NMR spectrometer (Bruker Biospin, Karlsruhe, Germany) operating at 600.13 MHz and equipped with an inverse detection 5 mm $^1$H-$^{13}$C-$^{15}$N-$^{31}$P cryoprobe connected to a cryo-platform and a cooled SampleJet sample changer, using Topspin v.3.2 software (Bruker, GmbH, Karlsruhe, Germany). $^1$H NMR spectra were recorded using a Carr-Purcell-Meiboom-Gill (CPMG) spin-echo pulse sequence to suppress broad signals from macromolecules and proteins. A water suppression was achieved by presaturation during the relaxation delay. The spin-echo loop time was adjusted to 240 ms, and a total of 512 free induction decays (FID) were acquired. Typical acquisition parameters included 32 k data points, a spectral width of 20 ppm, and a relaxation delay of 2 s. An exponential window function with a line-broadening factor of 0.3 Hz was applied to the FID before Fourier transformation. The spectra were manually corrected for phase and baseline distortions and referenced to TSP (δ 0 ppm) using Topspin v.3.5 software (Bruker Analytik, Rheinstetten, Germany). Metabolites were identified using an in-house database including reference spectra of metabolites recorded in the same conditions.

NMR spectra were imported in the AMIX software v.3.9, (Bruker, Rheinstetten, Germany) for data processing. A variable size bucketing was used based on graphical pattern and each integrated region was normalized to the total intensity.

## Analyzing the metabolic profile of *Lactobacillus* conditioned media through GC-MS profiling

To test what metabolites might be produced by *Lactobacillus* strains, we grew *L. rhamnosus*, *L. murinus* and a human isolate of *E. coli* in 5 ml of pre-reduced BHI media inside of the anaerobic chamber. After a 3-fold increase in absorbance, cultures were centrifuged for 3 min at maximum speed and the supernatants were passed through 0.22 µm filters and aliquoted into 3 tubes. As a control, non-inoculated BHI was also processed. Next, tubes were put in SpeedVac for 1 h at 30 °C after which pellets were stored at −80 °C until further processing. Pelleted metabolites were mixed with 50 µl of 2% methoxyamine hydrochloride (TS45950, Thermo Fisher) and incubated for 2 h at 30 °C with 1400 rpm shaking. Samples were further derivatized by adding 70 µl of ethyl acetate and 80 µl of *N*-Methyl-*N*-(trimethylsilyl)

trifluoroacetamide (MSTFA) + 1% TMCS (TS48915, Thermo Fisher) and were incubated at 37 °C for 30 min. Metabolites were analyzed on an Agilent 8890 GC coupled to Agilent 5977B mass selective detector. The GC was run in 1:20 split mode. A microliter of derivatized sample was injected onto a column and the GC oven temperature ramped from 60 °C to 325 °C. Peaks representing metabolites of interest were extracted using an in-house Matlab script. Fitlme function (implemented in Matlab R2021b) was used to determine which peaks were significantly different between tested conditions as compared to non-inoculated BHI and irrespectively of the possible variation between the replicates. Peaks of interest were matched against known spectra in NIST library.

## Proteomic and metagenomic analysis

The identification of bacterial non-redundant ORFs within human fecal samples and peptides in human fecal samples that match those ORFs was previously published[31]. In this study, the detected ORFs were translated into amino-acids and queried against the KEGG database[72]. Annotation was performed using HMMer v.3.1.2[73], with the following parameters: only Hits with an e-value lower than 0.05 and a minimum coverage of 0.50 were kept as significant results. For each ORF, only the best hit was kept.

## Statistical analysis

The Shapiro-Wilk test was applied in order to define if the populations under comparison followed a normal distribution. According to the results of this test, a non-parametric test (Mann–Whitney test, also known as Wilcoxon rank-sum test) or a parametric test (*T*-test) was applied in order to identify significant differences in specific features between two groups of samples. To identify significant differences in the number of CFUs, the Mann–Whitney test was applied since the populations did not follow a normal distribution. The T-test was applied to identify differences in the diversity between groups of samples since diversity levels followed a normal distribution.

To identify commensal bacteria in patients whose changes in abundance were associated with changes in MRE, we quantified the log2FC of the relative abundance of genera between consecutive samples collected from the same patient and correlated it with the log$_2$FC in MRE levels detected in the same pair of consecutive samples using the Spearman's correlation test. Only taxa that could be classified at least to the family level and with a median abundance >0.01% were included in this analysis (Supplementary Data file 3, abund). To adjust for multiple hypothesis testing and obtaining q values, we used the FDR approach by Benjamini and Hochberg[74], implemented in the p.adjust function of the R stats package v.3.6.0. An additional analysis was performed with taxa with a median abundance <0.01% (Supplementary Data file 3, low). In order to calculate the FC when the value in one of the samples was 0, a pseudo-count equal to the limit of detection was added to all analyzed samples.

To identify the commensal bacteria that better discriminate between untreated mice (resistance to MRKP colonization) and mice that were allowed to recover for 2 or 4 weeks before MRKP inoculation (susceptible to MRKP colonization) (Fig. 2d, Supplementary Data File 5), the linear discriminant analysis (LDA) effect size (LEfSe) was applied[75]. Only those taxa with a $p < 0.05$ and a LDA effect size > |2| were considered significantly different.

In order to analyze if MRE changes were associated with overall microbiota changes (Fig. 1b), a non-parametric test, permutational multivariate analysis of variance (PERMANOVA), was applied using the table of Euclidean distances obtained as described above and a vector containing the log2FC MRE values. The test was applied with the adonis function from the R vegan package v.2.5-3. The same test was applied to analyze if the microbiota of mice that recovered from antibiotic treatment was significantly different than the one of untreated mice (Fig. 2b). In this case the test was applied using the

table of Bray–Curtis distances between samples obtained as described above.

The Ancom2 test was applied in order to detect significant differences in the relative abundance of specific taxa between the groups of mice that did or did not receive *Lactobacillus* after antibiotic cessation (Fig. 4b, Supplementary Data File 6). This statistical test takes into account the compositional limitations to reduce the false discovery rate while maintaining high statistical robustness[76]. To adjust for multiple hypothesis testing, we used the FDR approach by Benjamini and Hochberg implemented in the p.adjust function of the R stats package v.3.6.0[74]. Only prevalent taxa (median > 0.01% in any of the groups under comparison) that could be classified to at least the family level were included in this analysis. Results from taxa that could be classified to the genus level are shown in figures (Fig. 4b). Results from taxa that could be classified to the genus or family level are shown in Supplementary Data File 6. The same analysis was performed to detect significant differences after the administration of the Clostridiales cocktail (Supplementary Fig. 11, Supplementary Data File 9).

In order to detect differences in the number of bacterial cells based on the quantification of 16S rRNA gene through qPCR (Supplementary Fig. 9A), the Mann–Whitney non-parametric test was applied. The same test was applied to identify significant differences in the levels of each taxon (Supplementary Fig. 9B, Supplementary Data File 7), which was inferred by multiplying the relative abundances obtained through 16S rRNA sequencing by the number of bacterial cells in that sample (obtained as described above). The same taxa previously included in the Ancom2 analysis were also included in this analysis. The Benjamini and Hochberg approach was used to adjust for multiple hypothesis testing.

In order to detect significant differences in the levels of metabolites between mice that did or did not receive *Lactobacillus* (Fig. 5a, Supplementary Fig. 13, Supplementary Data File 10), the non-parametric Mann–Whitney (Wilcoxon rank-sum test) was applied using the wilcox.test function in the "stats" R package. As described above, we used the Benjamini and Hochberg approach to adjust for multiple hypothesis testing. We first detected differences in the levels of metabolites by comparing the group of mice that received *Lactobacillus* (combining *L. murinus* and *L. rhamnosus*) to controls that didn't receive any strain which led to the identification of 17 metabolites whose abundance were significantly different (Supplementary Data File 10). Next, we applied the same statistical analysis on these 17 metabolites but separating the *Lactobacillus* group into two groups (the group of mice that received *L. murinus* and the group of mice that received *L. rhamnosus*) to identify those changes that were shared upon the administration of both *Lactobacillus*.

In order to detect associations between (i) metabolite levels and the relative abundance of bacterial taxa, (ii) levels of MRE and the relative abundance of bacterial taxa or metabolites, the Spearman correlation test was applied (Figs. 4c, 5b, f, and 6d; Supplementary Data Files 11 and 12). To adjust for multiple hypothesis testing, we used the Benjamini and Hochberg approach. Only prevalent taxa (median > 0.01%) that could be classified to at least the family level were included in this analysis. Results from taxa that could be classified to the genus levels are shown in figures (Figs. 4c, 5f, and 6d). Results for taxa that could be classified to the genus or family level are shown in Supplementary Data Files 11 and 12.

In order to study associations between the dynamics of *Lactobacillus* and the dynamics of different taxa in human samples, a Spearman correlation analysis was performed between the log2FC of the relative abundance of *Lactobacillus* between a pair of samples and the log2FC of the relative abundance of a particular taxon between the same pair of samples (Fig. 6d, Supplementary Data File 12). Like described above, we used the Benjamini and Hochberg approach to adjust for multiple hypothesis testing. Only prevalent bacteria (median > 0.01%) that could be classified to at least the family level were included in this analysis.

Results from taxa that could be classified to the genus levels are shown in figures (Fig. 6d). Results for taxa that could be classified to the genus or family level are shown in Supplementary Data File 12. Pairs of samples that do not contain any *Lactobacillus* count in both samples compromising the pair were not included in this analysis since we could not study how changes in *Lactobacillus* could influence the changes in other bacteria in these pairs of samples.

Associations between the levels of enzymes involved in production of butyrate (proteomic data) and butyrate levels or MRE levels (Fig. 6c, Supplementary Fig. 17), were performed using the Spearman correlation test as described above. In 45% of the analyzed samples, we were not able to detect peptides matching enzymes involved in production of butyrate. Since it is unlikely that these samples do not contain enzymes involved in production of butyrate and non-detection is likely a limitation of the sensitivity of the technique, these samples were excluded from the correlation analysis.

All tests applied were two-sided except when we evaluated the association between the log2FC in the relative abundance of OTUs from the genus *Lactobacillus* and the log2FC of MRE levels since we expected to identify OTUs that were negatively associated with MRE levels taking into account the results obtained at the genus level.

An analysis was considered statistically significant if the *p* value was lower than 0.05 and, in addition, the *q* value was lower than 0.1.

One single value (measurement) per sample has been included in all the analysis and figures.

### Reporting summary

Further information on research design is available in the Nature Research Reporting Summary linked to this article.

## Data availability

16S rRNA sequencing data has been deposited in the Sequence Read Archive (NCBI) under accession code numbers: PRJNA870718 for murine fecal samples, PRJNA870756 for human fecal samples. Metabolomic data from fecal samples has been deposited in the EMBL-EBI MetaboLights database[77], with the identifier MTBLS5811. Metabolomics data from *Lactobacillus* secretome experiments have been deposited to the EMBL-EBI MetaboLights with the identifier MTBLS5733. Proteomic data was previously published[31], and deposited in ProteomeXchange Consortium under accession number PXD011515, and the shotgun sequencing data for ORF identification has been deposited in the Sequence Read Archive (NCBI) under the accession number PRJNA877821. MRE levels and clinical patients data can be accessed in NCBI in the supplementary information of the previous published manuscript[17]. Tables containing the abundance of Genera, OTUs, KEGGs and metabolites from human samples are included as Supplementary material (Supplementary Data File 14). Tables containing the abundance of Genera and metabolites from murine fecal samples are also included as supplementary material in the Supplementary Data File 14. A table containing the abundances corresponding to metabolites detected in *Lactobacillus* secretome experiment is also included as Supplementary material in Supplementary Data file 14. The following databases/datasets have been used in this manuscript: 16S rRNA SILVA reference alignment implemented in Mothur (https://mothur.org/wiki/silva_reference_files/), Mothur-formatted RDP training set (https://mothur.org/wiki/rdp_reference_files/), NCBI RefSeq database (https://www.ncbi.nlm.nih.gov/refseq/), NIST library (https://www.nist.gov/programs-projects/tandem-mass-spectral-library), KEGG database (https://www.genome.jp/kegg/ko.html). We also used the DNA sequences of the NIBIOHN cohort, which have been deposited in DDBJ under accession numbers DRA010837– DRA010841 (https://ddbj.nig.ac.jp/DRASearch/study?acc=DRP007218, DRP007219, DRP007220, DRP007221, DRP007222). The DNA sequences of the MORINAGA cohort, which have been deposited in DDBJ under accession numbers

DRA009764 – DRA009767 (https://ddbj.nig.ac.jp/DRASearch/study?acc= DRP005906). The DNA sequences from the Flemish Gut Flora project sequence database (PMID: 30718848), Milieu Intérieur project sequence database (PMID: 31519223) and the Spanish Gut Microbiome Project sequence database (PMID: 34759297) are available upon request. Source data are provided with this paper.

## Code availability

Peaks representing metabolites of interest from the *Lactobacillus* conditioned media were extracted using Matlab R2021b and an in-house script available at https://github.com/anadj-micro/Analyzing-Lactobacillus-conditioned-media.git.

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

## Acknowledgements

C.U. was supported by the InfectERA-ERANET-Acciones complementarias grant [PCIN-2015-094] from the Spanish Ministerio de Economía y Competitividad and the 7th Research framework program from EU, grants from Conselleria d'Innovació, Universitats, Ciència i Societat Digital [AICO/2019/266, CIPROM/2021/053] and a grant from the Spanish MICINN [PID2020-120292RB-I00]. A.D. was supported by a Boehringer Ingelheim Fonds travel grant, and with J.B.X. by grant R01 AI137269/AI/NIAID from the US National Institutes of Health. B.K. was supported by the BMBF FloraStopMRE grant [031L0089]. V.C. was supported by an European Commission grant [MSCA-IF-2018-843183]. J.S. was supported by an InfectERA-ERANET-Acciones complementarias de programación conjunta internacional grant (AC15/00070) and Proyectos de Investigación en Salud del Instituto Carlos III (PI18/00280). K.B.X. was supported by Fundação para a Ciência e a Tecnologia grant (InfectERA/0004/2015). J.M.R. was supported by ANR FloraStopInfectMRE project. We thank the Metabohub-Metatoul platform for NMR analyses that were performed on instruments funded by the French National Infrastructure for Metabolomics and Fluxomics MetaboHUB (ANR-11-INBS-010). We thank A. Mira and A. López for sharing material required for the qPCR bacterial quantification. We thank E. Pamer for sharing the MRKP strain and the *Barnesiella* isolate. We thank J. Raes and the Flemish Gut Flora Project team for sharing their 16S sequencing database. We thank A. Latorre and M. Hernandez for sharing their 16S sequencing database of the Spanish Gut Microbiome with us. We thank Milieu Intérieur for sharing their 16S sequencing database with us.

## Author contributions

A.D., J.S., B.K., J.M.R., L.D., K.B.X., J.B.X., and C.U. designed research studies. A.D., M.J.G., V.C. performed mouse experiments. C.U. and K.B.X. supervised mouse experiments. A.D., I.P., and M.J.G. performed ex vivo and in vitro studies. A.D., M.J.G., C.C., and J.R. processed fecal samples to obtain omic data. C.U., A.D., M.J.G., C.C., M.G.G., J.R., M.T.F., L.P.C., and A.P.L. analyzed omic data. C.U., B.K., L.D., and J.B.X. supervised omic data analysis. E.M.G., M.S., J.L.L., M.A.S., and J.S. obtained clinical data and fecal samples. R.L. and A.D. obtained bacterial isolates. J.M.R. and C.U. supervised the obtention of bacterial isolates. A.D. and C.U. wrote the manuscript. All authors read and approved the final manuscript. J.S., B.K., J.M.R., L.D., K.B.X., and J.B.X. contributed equally to this manuscript.

## Competing interests

M.S. has participated as a consultant in advisories and scientific meetings organized or sponsored by the companies GSK, Gilead, MSD, Pfizer, and Shionogi in the last two years, having given lectures for some of them. There is no direct overlap between the current study and these consulting duties. C.U. has participated as a consultant of Vedanta Biosciences and The Zambon Group. There is no direct overlap between the current study and these consulting duties. The rest of the authors do not have competing interest.

## Additional information

¹Fundación para el Fomento de la Investigación Sanitaria y Biomédica de la Comunitat Valenciana - FISABIO, Valencia, Spain. ²Computational and Systems Biology Program, Sloan-Kettering Institute, New York, NY, USA. ³Toxalim – Research Center in Food Toxicology, Toulouse University, INRAE UMR 1331, ENVT, INP-Purpan, Paul Sabatier University, Toulouse, France. ⁴Instituto Gulbenkian de Ciência, Oeiras, Portugal. ⁵Aix Marseille Univ, IRD, APHM, MEPHI, IHU-Méditerranée Infection, Marseille, France. ⁶Department of Fundamental Microbiology, University of Lausanne, Lausanne, Switzerland. ⁷Chair of Proteomics and Bioanalytics, Technical University of Munich, Freising, Germany. ⁸Drug Discovery Unit, Instituto de Investigación Sanitaria La Fe, Valencia, Spain. ⁹Molecular Therapeutics Program, Centro de Investigación Médica Aplicada, University of Navarra, Pamplona, Spain. ¹⁰Hospital Universitari i Politècnic La Fe, Valencia, Spain. ¹¹Instituto de Investigación Sanitaria La Fe, Valencia, Spain. ¹²CIBERONC, Instituto Carlos III, Madrid, Spain. ¹³Centers of Biomedical Research Network (CIBER) in Epidemiology and Public Health, Madrid, Spain. ¹⁴These authors contributed equally: Cécile Canlet, Vitor Cabral, Rym Lalaoui, Marc García-Garcerá, Julia Rechenberger. ✉e-mail: ubeda_carmor@gva.es

