## [Peer Review File · Nature Communications]

Lactobacillus supports Clostridiales to restrict gut colonization by multidrug-resistant Enterobacteriaceae.REVIEWER COMMENTS

Reviewer #1 (Remarks to the Author):

The article by Djukovic and coworkers report that Lactobacilli contribute to resistance against multidrug-resistant Enterobacteriaceae (MRE). The presence of Lactobacilli correlates with increased levels of Clostridia and increased butyrate levels to protect against MRE.

Overall evaluation

The authors provide information relevant for an important nosocomial infection, but the findings are not novel (point 1) and remain largely correlative (point 2). As a result, the study marks an incremental advance.

Major points

1) The authors find that Lactobacilli contribute to colonization resistance against Enterobacteriaceae, but this is confirmatory to a 1988 report by Itoh and Freter (reference 38). This study also reported in its title on the “control of Escherichia coli populations by a combination of indigenous Clostridia and Lactobacilli in gnotobiotic mice”, thus the observation of the authors that the mechanism of colonization resistance involves cooperation between Lactobacilli and Clostridia is also confirmatory. The authors go on to suggest that colonization resistance is due to production of butyrate by Clostridia and nutrient depletion by Clostridia. The fact that short chain fatty acids, such as butyrate, contribute to growth inhibition of Enterobacteriaceae has been reported extensively, starting with seminal work by Meynell in the 1960's. Depletion of amino acids by Clostridia as a mechanism for colonization resistance against Enterobacteriaceae has also been reported recently (PMID: 32726577). Thus the mechanism of colonization resistance proposed by the authors is also confirmatory.

2) Considering that cooperation between indigenous Clostridia and Lactobacilli in conferring colonization resistance and mechanisms by which Clostridia contribute to colonization resistance against E. coli have already been described, the authors would need to break new ground to move the field forward. One path forward would be to determine whether Lactobacilli are responsible for enhancing Clostridia recovery after antibiotic treatment (causality) by elucidating the underlying mechanism. Answering this question would add novelty.

Reviewer #2 (Remarks to the Author):

Overall, this is an exciting study led by Garzon et al focuses on the role of Lactobacillus in protecting the gut from drug resistant Klebsiella infection, and how this is mediated through metabolites that support Clostridia and deter Klebsiella. The premise of exploiting interactions and not just individual microbes for understand gut microbiome infection protection mechanisms is clearly critical, especially in an era where individualized probiotics are being marketed without evidence to support them. I suggest publication after attention to the following comments:

1. Abstract: please add some information about study design and number of people included, along with a bit of specificity about which Lactobacillus is key, and describe what kind of interaction/cooperation between Lacto and Clostridia is involved
2. Line 45, 46, 47 have dashes that are awkwardly spaced
3. Line 99 and description of Lacto as key: is it possible to be more quantitative about the idea that Lacto is key but not sufficient?
4. Line 137 and throughout: Please add R2 values to Permanovas and describe the variance explained
5. Line 171 - was this Lacto OTU prevalent across many patient samples? Is it common outside of this cohort? How does it compare to Lactos sold as probiotics? The recent renaming of several Lactos makes the species designation especially important
6. Line 171 suggests to go to the methods, and then in the results it is suggested to go to the methods (later there is a bit more info), however, please clarify and give a synopsis of the information from the earlier paper
7. Figure 1. Please label the columns In Fig 1A (do the circles each represent different patients in the x-axis/column direction?)
8. Please label x-axes in Figure 2; hard to follow what is shown in Fig 2E (legend is there, but words blend together a bit)
10. Line 1105 extra word "in"
11. Combining the two Lacto strains in Figure 4 is confusing - with these two strains, it feels hard to conclude something related to all Lactos. Perhaps add clarification in the Figure caption title or in the figure label
12. Similar to the generalization of the Lactos, how do the Clostridia chosen for this study relate to sequenced human and murine Clostridia? How do they relate to typically pathogen C diff? Could Lacto potentially also protect against C diff?

13. Fig 6 - nice inclusion of comparative patient data

14. Line 296 - great inclusion of pH control - how was this done? (Lacto could induce steep local pH gradients, this might be an important caveat)

REVIEWER COMMENTS

Reviewer #1 (Remarks to the Author):

The article by Djukovic and coworkers report that *Lactobacilli* contribute to resistance against multidrug-resistant *Enterobacteriaceae* (MRE). The presence of *Lactobacilli* correlates with increased levels of *Clostridia* and increased butyrate levels to protect against MRE.

Overall evaluation

The authors provide information relevant for an important nosocomial infection, but the findings are not novel (point 1) and remain largely correlative (point 2). As a result, the study marks an incremental advance.

Major points

1) The authors find that *Lactobacilli* contribute to colonization resistance against *Enterobacteriaceae*, but this is confirmatory to a 1988 report by Itoh and Freter (reference 38). This study also reported in its title on the “control of *Escherichia coli* populations by a combination of indigenous *Clostridia* and *Lactobacilli* in gnotobiotic mice”, thus the observation of the authors that the mechanism of colonization resistance involves cooperation between *Lactobacilli* and *Clostridia* is also confirmatory. The authors go on to suggest that colonization resistance is due to production of butyrate by *Clostridia* and nutrient depletion by *Clostridia*. The fact that short chain fatty acids, such as butyrate, contribute to growth inhibition of *Enterobacteriaceae* has been reported extensively, starting with seminal work by Meynell in the 1960’s. Depletion of amino acids by *Clostridia* as a mechanism for colonization resistance against *Enterobacteriaceae* has also been reported recently (PMID: 32726577). Thus the mechanism of colonization resistance proposed by the authors is also confirmatory.

We appreciate the reviewer comment indicating that our manuscript provides relevant information about an important nosocomial infection, which concords with the opinion of the second reviewer. However, we disagree about the lack of novelty of our manuscript and that our observations remain largely correlative:

(i) The study from Itoh and Freter showed that inoculation of *Lactobacillus* and a mixture of chloroform resistant bacteria (supposedly spore forming *Clostridia*, whose spores are resistant to this treatment) decrease the intestinal levels of *E. coli* in germ free mice to a higher extent than when each of them were inoculated separately. However, the exact composition of the bacteria used in these studies is unclear. The authors assume that mice were exclusively colonized with *Clostridia* and *Lactobacillus*. However, due to the inability to analyze the composition of the microbiota at that time, this was not confirmed. Thus, it is

highly possible that other bacteria known to produce spores (e.g. Erysipelotrichales) may have also been administered to mice and could have contributed to the observed phenotype. In other words, the results obtained by that study, although important, were not conclusive since appropriate techniques to analyze the microbiota of mice were not available at that time. In contrast, in our study we colonized the mice with pure cultures of *Lactobacillus* and Clostridiales strains and confirmed the murine intestinal colonization of the administered bacteria through 16s rRNA sequencing.

(ii) Importantly, our data pinpoints the mechanism by which *Lactobacillus* interacts with Clostridiales to suppress multidrug-resistant Enterobacteriaceae (MRE) intestinal colonization: *Lactobacillus* promotes the expansion of Clostridiales, which then inhibit MRE intestinal colonization through creating a detrimental environment for MRE growth. This cooperative mechanism was not described in the mentioned manuscript in which the authors only evaluated the effect of inoculating both types of bacteria on the levels of *E. coli* colonization in GFM, without providing potential mechanisms that could explain the synergy between both groups of bacteria. Moreover, as discussed later, our results were validated using two different mouse models (GFM, as in the mentioned manuscript, and antibiotic-treated mice), two different *Lactobacillus* strains (murine and human origin) and a human cohort of patients highly susceptible to MRE infections.

(iii) In our study we used a mouse model that mimics the conditions of hospitalized patients (i.e. treatment with multiple antibiotics) in order to demonstrate that the mechanism by which *Lactobacillus* restricts MRE intestinal colonization is by promoting the expansion of Clostridiales which can *per se* reduce the intestinal colonization of MRE. This result is novel, and this type of cooperative mechanism (*Lactobacillus* promoting the recovery of Clostridiales after antibiotic treatment), as mentioned earlier, was not proposed in the study of Freter et al. We do not think, as the reviewer suggested, that this result is just correlative, but rather we think that with the exhaustive experiments that we have performed we have demonstrated that *Lactobacillus* is the cause of the Clostridiales expansion which in turn is the cause of the reduction of MRE levels. This was demonstrated in multiple reproducible experiments by showing that oral inoculation of *Lactobacillus* to antibiotic treated mice increases the recovery of Clostridiales taxa. This result was confirmed with two different *Lactobacillus* strains from different origin (mouse and human). Moreover, as opposed to the Itoh and Freter study, in which the colonization levels of Clostridiales bacteria were not evaluated, we performed quantitative PCR in order to confirm the expansion in absolute numbers of Clostridiales. Subsequently, we isolated and administered Clostridiales strains (not a mixture of uncharacterized chloroform resistant bacteria) to antibiotic-treated mice to show that in the absence of *Lactobacillus*, Clostridiales are capable of restricting MRE intestinal colonization. This result in combination with results in GFM in which *Lactobacillus per se* is not able to restrict MRE intestinal colonization indicated that the mechanism by which *Lactobacillus* confers protection is by facilitating the recovery of Clostridiales after antibiotic administration, a novel mechanism not shown in the Freter et al study.

(iv) Importantly, the mouse experimental data was combined with omic analysis of human fecal samples in order to show that a similar cooperation between *Lactobacillus* and Clostridiales occurs in hospitalized patients to restrict MRE intestinal colonization. A very relevant and novel result that had not been previously addressed due to the lack of clinical studies analyzing the levels of MRE in combination with metagenomics, metabolomics and proteomics data in hospitalized patients. In fact, to our understanding and thanks to the addition of the human data, our results propose for the first time a cooperative mechanism between two major commensal bacteria that restricts gut colonization by a pathogen both in mice and in the human population.

(v) Another relevant difference in our study, specially related to the translation of results into the clinics, is that all our results, both in patients and in the mouse model, are derived from the study of Enterobacteriaceae multidrug resistant clinical strains, rather than the utilization of model bacteria, as used in other studies. This is relevant because similar mechanisms of colonization resistance could have a different impact depending on the studied strain. And therefore, it is important to identify mechanisms by which commensal bacteria restrict colonization by those pathogenic strains that are causing clinical problems in susceptible patients, as we do here.

(vi) We agree with the reviewer that using mouse models it has previously been shown that Clostridiales restrict Enterobacteriaceae colonization by production of butyrate. However, this has never been shown in the specific context of this manuscript: administration of *Lactobacillus* and subsequent expansion of Clostridiales after antibiotic treatment. This context is relevant because it mimics what could be happening in hospitalized patients at risk of developing infections with MRE. Moreover, by performing metabolomic and proteomic analysis of the clinical samples, we were able to show that butyrate and enzymes related to butyrate production are associated with a decrease in MRE levels, suggesting for the first time that butyrate production by Clostridiales is key for restricting MRE intestinal colonization also in the human population.

2) Considering that cooperation between indigenous Clostridia and Lactobacilli in conferring colonization resistance and mechanisms by which Clostridia contribute to colonization resistance against *E. coli* have already been described, the authors would need to break new ground to move the field forward. One path forward would be to determine whether Lactobacilli are responsible for enhancing Clostridia recovery after antibiotic treatment (causality) by elucidating the underlying mechanism. Answering this question would add novelty.

As indicated above, with the experiments performed, we believe that we have demonstrated a causal role for *Lactobacillus* in promoting the recovery of Clostridiales after antibiotic administration. This was demonstrated in reproducible experiments by administering either *L. murinus* or *L. rhamnosus* and

analyzing the levels of Clostridiales bacteria as compared to a proper control group of mice that did not receive *Lactobacillus*. Thus our conclusions about the role of *Lactobacillus* in promoting Clostridiales expansion are not derived from correlation analysis but from *in vivo* experimental data.

We agree that it would be very interesting to identify the specific mechanism by which *Lactobacillus* boost Clostridiales. One possibility, as suggested in the discussion, would be that *Lactobacillus* metabolism derived products can directly boost Clostridiales growth. As a proof of concept, through GS-MS, we identified products specifically derived from *Lactobacillus in vitro* growth (i.e. lactate, ribose and adenine; see below new Suppl. Fig. 10 and lines 258-261 of the results section and 445-451 of the discussion section in the manuscript). Next, we evaluated if these metabolites could promote the growth of Clostridiales. To this end we used as a model organism a strain of *Flavonifractor spp*, one of the Clostridiales that expands after *Lactobacillus* administration in mice, that is able to grow in BHI (the media in which we grew *Lactobacillus in vitro*). Notably, *Flavonifractor* growth was significantly enhanced when the media was supplemented with the 3 metabolites produced by *Lactobacillus* (new Suppl. Fig. 10 included in the manuscript). This experiment demonstrates that *Lactobacillus* strains are able to produce metabolites that can boost the growth of certain Clostridiales strains, which supports a potential mechanism by which *Lactobacillus* could be facilitating the expansion of Clostridiales. We acknowledge that verifying this potential mechanism, in a comprehensive manner as we have defined the role of *Lactobacillus* and Clostridiales in restricting MRE intestinal colonization, would require extensive additional work. This would include the generation of *Lactobacillus* mutants defective in specific metabolic pathways, *ex vivo* experiments combined with mouse models and multiomic human data analysis. Thus the complete confirmation of the mechanism by which *Lactobacillus* promotes Clostridiales growth in mice and humans represents a novel project that we consider to be out of the scope of this work and that we expect to develop after publication of this study.

B ■ BHI ■ BHI + metabolites

Supplementary Figure 10. *Lactobacillus* derived metabolites promote the growth of a Clostridiales strain *in vitro*. *L. murinus*, *L. rhamnosus* or *E. coli* were grown in BHI. **(A)** GC-MS identified metabolites whose abundance changes (log₂FC) in *Lactobacillus* conditioned media, but not in *E. coli* conditioned media ($p < 0.05$ as compared to the abundance of that particular metabolite in BHI media). Metabolites of interest (those that increased upon *Lactobacillus* growth) are indicated. **(B)** *Flavonifractor plautii* was grown in BHI or BHI supplemented with metabolites identified in (A): adenine, ribose, lactate. **(C)** Area under the curve of the growth curve shown in (B). The growth of *F. plautii* was significantly increased by the administration of metabolites derived from *Lactobacillus* growth (**** $p < 0.0001$; Student's T-Test).

We believe that the conclusions from this study, which are independent on the complete demonstration of the specific mechanism by which *Lactobacillus* boost Clostridiales, have already the strength, novelty and relevance standards for the broad audience of Nature Communications:

(i) We have identified specific members of the microbiome (*Lactobacillus* and Clostridiales) key for restricting multidrug-resistant Enterobacteriaceae **in both**

mice and patients.

(ii) We have identified **in the human population** specific mechanisms of protection against MRE using a combination of multiple omic techniques, including metagenomics, metabolomics and proteomics (i.e. butyrate production).

(iii) We have identified **in both mice and humans** a cooperative mechanism by which the interaction between two major members of the microbiome restricts the intestinal colonization by MRE.

To our knowledge this is the **first example of cooperation among members of the microbiome shown to confer protection against pathogens in humans.**

However, considering the microbiome diversity and metabolic dependence among its members, it is very likely that other unknown interactions between commensals may be taking place to confer protection against MRE and also against other pathogens in patients. Our results therefore will be inspiring for the scientific community and will open new lines of research to find such interactions. As the second reviewer indicated, exploiting the novel identified bacterial interactions and not just individual microbes will be critical for developing novel probiotics to treat infections. For all these reasons, we would not like to delay the publication of these important results, which we think the scientific and clinical community should be aware of before the identification of the exact molecular mechanism by which *Lactobacillus* boost Clostridiales growth.

Reviewer #2 (Remarks to the Author):

Overall, this is an exciting study led by Garzon et al focuses on the role of Lactobacillus in protecting the gut from drug resistant Klebsiella infection, and how this is mediated through metabolites that support Clostridia and deter Klebsiella. The premise of exploiting interactions and not just individual microbes for understand gut microbiome infection protection mechanisms is clearly critical, especially in an era where individualized probiotics are being marketed without evidence to support them. I suggest publication after attention to the following comments:

We appreciate that the reviewer finds our manuscript exiting and that considers that the manuscript could be appropriate for publication in this journal after considering her/his comments. Below we reply to all the reviewer's comments which we think have improved the manuscript. We hope that now the manuscript will be suitable for publication in Nature Communications.

1. Abstract: please add some information about study design and number of people included, along with a bit of specificity about which Lactobacillus is key, and describe what kind of interaction/cooperation between Lacto and Clostridia is involved.

We have included more details into the abstract regarding the number of patients, type of study and indicated the species of *Lactobacillus* tested in the manuscript.

We cannot include more details since the abstract has a very strict word limitation (maximum 150 words). Thus we could not include most of the details regarding the study designed that were included in the methodology section.

2. Line 45, 46, 47 have dashes that are awkwardly spaced

We have replaced the dashes by commas which are more appropriate.

3. Line 99 and description of Lacto as key: is it possible to be more quantitative about the idea that Lacto is key but not sufficient?

We have modified the sentence so that now we described exactly to what extent *Lactobacillus* restricts MRE intestinal colonization:

*“Next, using a mouse model we demonstrated that *Lactobacillus* sp. is required but not sufficient to restrict MRE intestinal colonization (i.e. two orders of magnitude lower MRE levels compared to control mice).”*

4. Line 137 and throughout: Please add R2 values to Permanovas and describe the variance explained

We have added the R2 values, next to the p values in the text, as a measure of the variance explained.

5. Line 171 - was this Lacto OTU prevalent across many patient samples?

The *Lactobacillus* OTU was present in the majority of the patients: 68% of the patients that were included in the analysis. We have added this information in the results section:

*“In addition, this OTU, being the most prevalent one within this genus (**Suppl. Table 3**), was present in 35% of the samples (68% of patients) as compared to the genus *Lactobacillus* that is present in 72% of the samples (93% of patients), suggesting that the negative association between *Lactobacillus* and MRE might be due to more than one OTU.”*

Nevertheless, we would like to pinpoint, as indicated in the results section, that the highest negative association detected with MRE levels was at the genus level: *Lactobacillus* ($\rho=-0.27$, $p=0.04$), rather than at the OTU level: *Lactobacillus* OTU20 ($\rho=-0.187$, $p=0.026$). Moreover, experiments in mice demonstrated that not only the *Lactobacillus* OTU20 isolate but also the *Lactobacillus* murine isolate were effective at restricting MRE intestinal colonization. And that both strains inhibit MRE intestinal colonization through promoting the expansion of Clostridiales bacteria. Thus rather than the role of a specific *Lactobacillus* OTU or strain on restricting MRE intestinal colonization, our results suggest that multiple *Lactobacillus* species may be able to suppress MRE intestinal colonization by promoting the expansion of Clostridiales species. For this reason,

we have not emphasized in the manuscript the role of *Lactobacillus* OTU20 and/or *L. rhamnosus* on protection against MRE intestinal colonization. In contrast, we intend to give a more general message about the role of different *Lactobacillus* species on suppressing MRE intestinal colonization through promoting the expansion of Clostridiales.

That said, we agree with the reviewer that it will be interesting expanding our knowledge about the prevalence of the OTU20, considering that it is the most prevalent *Lactobacillus* OTU in our cohort and that an isolate of the OTU20 (*L. rhamnosus*) was used as a model to study the protective role of *Lactobacillus* on MRE intestinal colonization. Thus we have performed all the analysis suggested by the reviewer, that have expanded our knowledge about the prevalence of this particular OTU in other cohorts and its phylogenetic relationship with other *L. rhamnosus* strains (see below).

Is it common outside of this cohort?

We have analyzed the presence of the *Lactobacillus* OTU20 in 5 additional cohorts of healthy individuals from two different continents (Europe and Asia) and four different countries (Japan, Belgium, France and Spain) containing a total of 4538 individuals. The *Lactobacillus* OTU was present in all analyzed cohorts, although its prevalence varied depending on the cohort (new Supplementary Fig. 3). We have included this information in a new Supplementary Figure and also in the results section:

“In order to demonstrate the protective role of Lactobacillus against MRE intestinal colonization, we first isolated a representative of the most prevalent Lactobacillus OTU from patient samples (see methodology for isolation in the methods section). This OTU is also present in other cohorts of healthy individuals of different continents (Suppl.Fig.3).”

Supplementary Figure 3. Prevalence of the *Lactobacillus* OTU20 on different human cohorts. A representative sequence of the *Lactobacillus* OTU20 was aligned against 16S rRNA sequences from different cohorts of different studies. The % of individuals containing sequences that could be assigned to the *Lactobacillus* OTU20 from our cohort (see methods) is indicated. N= Number of individuals included in the cohort. NA: only one cohort was included in that study and no specific name was given to that cohort.

How does it compare to Lactos sold as probiotics?

ATCC

DSMZ

The recent renaming of several Lactos makes the species designation especially important.

We have compared the genome sequence of our isolate against the genome sequences of 198 *Lactobacillus rhamnosus* (now renamed *Lacticaseibacillus rhamnosus*) that were deposited in NCBI. This include human isolates that have been tested as probiotics in different studies (red bars in the figure). As a measure of similarity we calculated the average nucleotide identity (ANI) between our isolate and all other genome sequences deposited in NCBI. As expected, our isolate is very similar to *L. rhamnosus* strains (average ANI = 98.7%), thus confirming that our isolate belongs to the species *L. rhamnosus*. However, the genome of our isolate was not identical to any of the strains analyzed and was not more similar to any particular human strain that has been tested as a probiotic. We have included these results in a new supplementary figure and in the results section:

“Whole genome sequencing (WGS) confirmed that the isolate belongs to the genus Lactobacillus, more precisely species Lactobacillus rhamnosus (now renamed to Lacticaseibacillus rhamnosus) (Suppl. Fig. 4; 98.7% identity). Extended analysis of our isolate as compare to L. rhamnosus genomes deposited in NCBI (N=198), including some used as probiotics, indicated that none of the sequenced strains is 100% identical to the strain isolated in our study (Suppl.Fig.5; Suppl. Table 4).”

Supplementary Figure 5. Genome similarity of the *L. rhamnosus* isolate obtained in this study compare to other *L. rhamnosus* isolates. (A) Phylogenetic tree was constructed with the core-genome of 198 *L. rhamnosus* genomes deposited in NCBI plus the *L. rhamnosus* isolate from this study (blue). **(B)** Average nucleotide identity between the *L. rhamnosus* isolated in this study and *L. rhamnosus* genomes. Red bars and fonts indicate *L. rhamnosus* strains that have been tested as probiotics in humans (see Supplementary Table 4).

6. Line 171 suggests to go to the methods, and then in the results it is suggested to go to the methods (later there is a bit more info), however, please clarify and give a synopsis of the information from the earlier paper

In line 171 we indicated that “in order to demonstrate the protective role of *Lactobacillus* against MRE intestinal colonization, we first isolated a representative of the most prevalent *Lactobacillus* OTU from patient samples (see methods)”.

We included in the methods the methodology to isolate the *Lactobacillus* OTU.

We did not include any information from the previous cited manuscript since in the previous manuscript we did not perform 16s rRNA sequencing analysis. In the previous manuscript we quantified the level of MRE in fecal samples, characterized their taxonomy and antibiotic resistance pattern and analyze how different clinical factors (antibiotics and antifungals received, neutropenia, mucositis, parenteral feeding) were associated with MRE levels. We have included these details in the methodology section:

“In the previous published study¹⁶, we analyzed the levels of MRE colonization in fecal samples from acute leukemia patients, characterized their taxonomy and resistance to antibiotics and investigate the associations between MRE levels and different clinical variables, including antibiotics and antifungals received, neutropenia, mucositis and parenteral feeding¹⁶. We demonstrated that beta-lactam administration significantly reduced the fecal levels of MRE strains sensitive to the administered beta-lactam¹⁶”.

We understand that the sentence written in the results section could be confusing and we have modified it to indicate that the methods in that particular sentence are referred to the methodology of how *L. rhamnosus* was isolated:

“In order to demonstrate the protective role of Lactobacillus against MRE intestinal colonization, we first isolated a representative of the most prevalent Lactobacillus OTU from patient samples (see methodology for isolation in the methods section)”.

7. Figure 1. Please label the columns In Fig 1A (do the circles each represent different patients in the x-axis/column direction?)

Figure 1A, as indicated in the figure legend, shows the data from one patient as an example of the data used in this manuscript to identify commensal bacteria associated with MRE colonization levels. The X-axis/column direction represents the different timepoints of each of the samples analyzed from this particular patient. This information has now been added to the figure and also clarified in the figure legend.

8. Please label x-axes in Figure 2; hard to follow what is shown in Fig 2E (legend is there, but words blend together a bit)

We have labeled the x-axes in the new version of the figure and modified the figure legend to make it more clear.

10. Line 1105 extra word “in”

Thanks for noticing this mistake which has been corrected.

11. Combining the two Lacto strains in Figure 4 is confusing - with these

two strains, it feels hard to conclude something related to all Lactos. Perhaps add clarification in the Figure caption title or in the figure label

We have indicated in the figure label that the data from the group that received *Lactobacillus* combines data from mice that received either *L. rhamnosus* or *L. murinus*. This was also indicated in the Figure legend. In addition, in the figure legend is indicated that separate results for mice that received either *L. rhamnosus* or *L. murinus* are shown in Supplementary Figure 8.

12. Similar to the generalization of the Lactos,

how do the Clostridia chosen for this study relate to sequenced human and murine Clostridia?

How do they relate to typically pathogen C diff?

Commensal bacteria isolated from humans were first identified through MALDI-TOF and later on the 16S sequence was compared to the NCBI database to identify the potential bacterial species. 16S rRNA sequence confirmed that the isolated *Blautia* is 99.73% identical to the strain DSM2950 from human origin *Blautia producta*, a commensal species frequently found in human fecal samples (Gupta V., *Nature Communications*, 2020) that has anti-inflammatory properties in vitro (Liu X, *Microorganism*, 2021). The isolated *Flavonifractor* is 99.57% identical to the human strain 265 from the species *Flavonifractor plautii*, a species that is also frequently found in human fecal samples (Gupta V., *Nature Communications*, 2020) and that has been shown to provide beneficial anti-inflammatory effects in mouse models (Roses C., et al., *Nutrients*, 2021; Mikami A et al., *Frontiers in Nutrition* 2021; Mikami A et al., *Molecular Biology Reports*, 2020). We have included the information of the species level in the new version of the manuscript.

“Subsequently, 16s rRNA PCR sequencing and comparison with the NCBI Reference RNA sequence database was applied to confirm the taxonomy of the isolated strains. By doing this procedure, the strains CU864 (Flavonifractor plautii) CU826 (Blautia producta) and CU700 (L. rhamnosus) were isolated.

The closest sequence to the murine *Oscillibacter* isolate in NCBI is the strain *Oscillibacter valericigenes*. Surprisingly, this is a strain isolated from clams. Nevertheless, the similarity with the 16s rRNA from this strain is below 97% (i.e. 95.29%), the threshold used to define the species level, suggesting that our isolate does not belong to the *O. valericigenes* species. It is possible that *Oscillibacter* species have not been well characterized in murine samples and this could be the reason why we were not able to identify any close related bacterial species in the NCBI database. To define if close related bacteria to our isolate were present in other murine colonies, we compared the sequence of our isolate against 16S sequences obtained from a study in which the murine microbiota of six different strains of mice (A/J, BALB/cJ, C57BL/6J, A/JOlA^{Hsd}, BALB/cAnNHsd, and C57BL/6NHs; N=8 mice per colony) from two different vendors (Jackson and Harlan) was analyzed through 16s rRNA sequencing

(Ericsson AC et al., PLoS One, 2015, 10.1371/journal.pone.0116704). Sequences compatible with being the same OTU as the *Oscillibacter* isolate from our colony (>97% identity and 100% coverage) were identified in all analyzed mice. The same result was obtained for the other murine isolate used in our study (Unclassified Ruminococaceae). These results suggest that close-related bacteria to the murine Clostridiales isolates used in our study are frequently present in murine colonies frequently used in laboratories worldwide.

Regarding the potential relation of our Clostridiales isolates to *Clostridioides difficile*, the similarity of 16S sequences to *C. difficile* was in all cases lower than 85%, suggesting that our isolates are phylogenetically distinct from the pathogen *C. difficile*.

Could Lacto potentially also protect against C diff?

This is an interesting question that we have not evaluated in the manuscript. We focused on multidrug resistant Enterobacteriaceae because it is a relevant pathogen that is frequently detected in our patient cohort but it will be interesting to perform future studies, both in mice and humans to evaluate if *Lactobacillus* could confer protection against *C. difficile* by promoting the expansion of other Clostridiales bacteria that may compete for similar nutrient sources. Regarding this, a recently published manuscript evaluated the effect of the administration of a 3-strain *Lactobacillus* probiotic cocktail on the prevention of *C. difficile* infections. Interestingly they found that administration of the probiotics significantly diminished the number of *C. difficile* infection cases. We have added this information in the discussion section:

“In addition, future studies should evaluate if the interaction of Lactobacillus and Clostridiales is involved in protection against other pathogens besides MRE. In this sense, a recently published study identified that the administration of a probiotic cocktail containing 3 Lactobacillus strains diminished the rate of Clostridioides difficile infection in hospitalized patients⁴³. It will be interesting to evaluate if in this case Lactobacillus can directly inhibit C. difficile or rather the expansion of Clostridiales species are responsible for this beneficial effect. For example, by promoting the expansion of Clostridium scindens, a Clostridiales commensal that can inhibits C. difficile through secondary bile acid production²⁰”.

13. Fig 6 - nice inclusion of comparative patient data

We thank the reviewer for this comment. We agree with the reviewer that the patient's data is an important addition to the results obtained from mice and adds novelty and importance to our study.

14. Line 296 - great inclusion of pH control - how was this done? (Lacto could induce steep local pH gradients, this might be an important caveat)

The measurement of pH was performed using a pH electrode that was introduced into the caecal content of mice. Thus we calculated the overall pH of the caecum. It is possible that pH gradients are generated within the intestinal contents that cannot be measured using the pH electrode. Thus future studies will be required to investigate to what extent local pH gradients could contribute to the inhibition of MRE. Nevertheless, considering the results obtained in germ free mice in which *Lactobacillus* was not able to restrict MRE intestinal colonization, potential local pH gradients generated by *Lactobacillus* do not seem to be sufficient to diminish MRE intestinal colonization levels.

REVIEWERS' COMMENTS

Reviewer #1 (Remarks to the Author):

The article by Djukovic and coworkers report that Lactobacilli contribute to resistance against multidrug-resistant Enterobacteriaceae (MRE). The presence of Lactobacilli correlates with increased levels of Clostridia and increased butyrate levels to protect against MRE.

Overall evaluation

The main weaknesses identified in a previous review were (i) that the findings are not novel, and (ii) that the findings remain largely correlative. Unfortunately, neither of these concerns were addressed experimentally and as a result, the study still marks an incremental advance.

Major points

1) Novelty: The main conclusion that Lactobacilli contribute to colonization resistance by cooperating with Clostridia to produce short-chain fatty acids and deplete nutrients is confirmatory. Specifically:

1.a) The finding that Lactobacilli contribute to colonization resistance against Enterobacteriaceae is confirmatory to reference 39.

1.b) The observation that the mechanism of colonization resistance involves cooperation between Lactobacilli and Clostridia is confirmatory to reference 39.

1.c) The finding that short chain fatty acids, such as butyrate, contribute to growth inhibition of Enterobacteriaceae is confirmatory to a large body of work, initiated by Meynell in the 1960's.

1.d) The finding that depletion of amino acids by Clostridia contributes to colonization resistance against Enterobacteriaceae is confirmatory to a previous report (PMID: 32726577).

The authors argue in their rebuttal letter that these four findings (points 1.a-d) had never been shown side-by-side in the same manuscript and that the authors used the latest technology and relevant isolates to demonstrate these findings, which provides novelty. However, these novel aspects represent an incremental advance conceptually.

2) Considering that cooperation between indigenous Clostridia and Lactobacilli in conferring colonization resistance and mechanisms by which Clostridia contribute to colonization resistance against *E. coli* have already been described, the authors would need to break new ground to move the field

forward. We suggested that one path forward would be to determine whether Lactobacilli are responsible for enhancing Clostridia recovery after antibiotic treatment (causality) by elucidating the underlying mechanism. The authors argue that these experiments are not necessary. As a result, the study still marks an incremental advance.

Reviewer #2 (Remarks to the Author):

I am very impressed by the authors thorough and attentive response to the reviewer comments, complete with extensive analysis of public data. The manuscript was already strong and now has a new level of depth. I really enjoyed seeing how the Lacto OTU was found in other cohorts, and how similar the sequence is to commercial probiotics, for example.

My only minor comment is that there are somewhat pervasive and small grammatical errors - the wrong prepositions are used in several places, for example. I advise having a layer of editing for correctness in English.

REVIEWERS' COMMENTS

Reviewer #1 (Remarks to the Author):

The article by Djukovic and coworkers report that *Lactobacilli* contribute to resistance against multidrug-resistant *Enterobacteriaceae* (MRE). The presence of *Lactobacilli* correlates with increased levels of *Clostridia* and increased butyrate levels to protect against MRE.

Overall evaluation

The main weaknesses identified in a previous review were (i) that the findings are not novel, and (ii) that the findings remain largely correlative. Unfortunately, neither of these concerns were addressed experimentally and as a result, the study still marks an incremental advance.

Major points

1) Novelty: The main conclusion that *Lactobacilli* contribute to colonization resistance by cooperating with *Clostridia* to produce short-chain fatty acids and deplete nutrients is confirmatory. Specifically:

1.a) The finding that *Lactobacilli* contribute to colonization resistance against *Enterobacteriaceae* is confirmatory to reference 39.

1.b) The observation that the mechanism of colonization resistance involves cooperation between *Lactobacilli* and *Clostridia* is confirmatory to reference 39.

1.c) The finding that short chain fatty acids, such as butyrate, contribute to growth inhibition of *Enterobacteriaceae* is confirmatory to a large body of work, initiated by Meynell in the 1960's.

1.d) The finding that depletion of amino acids by *Clostridia* contributes to colonization resistance against *Enterobacteriaceae* is confirmatory to a previous report (PMID: 32726577).

The authors argue in their rebuttal letter that these four findings (points 1.a-d) had never been shown side-by-side in the same manuscript and that the authors used the latest technology and relevant isolates to demonstrate these findings, which provides novelty. However, these novel aspects represent an incremental advance conceptually.

We acknowledge that previously published studies have provided data regarding the effect of *Lactobacillus*, *Clostridiales* and/or butyrate in colonization resistance against *Enterobacteriaceae*. Indeed, we cite studies mentioned by the reviewer and discuss our results accordingly.

However, we disagree with the reviewer regarding the level of incremental advance provided by our manuscript, and in agreement with the second reviewer, we think that the results from our work are highly novel and relevant, which makes them suitable for the broad audience of *Nature Communications*. Specifically, our manuscript provides the following advances to the field:

- i) We demonstrate that *Lactobacillus* cooperate with *Clostridiales* to suppress intestinal colonization by multidrug-resistant *Enterobacteriaceae* (MRE). Previous results indicated by the reviewer (ref. 39, now ref 40) suggested a potential cooperation of these two types of commensals against *Enterobacteriaceae* in germ-free mice (GFM). However, their results were not conclusive due to the (i) inability to verify the commensal bacteria administered to mice, (ii) inability to analyze the bacteria that colonize the intestine of mice. In contrast, in our study we colonize mice with defined bacterial isolates, including two different types of *Lactobacillus* strains (human and mouse origin) and we used metagenomic analysis to track colonization of the administered and endogenous commensal bacteria in mice. Moreover, we used two different mouse models to demonstrate

our results, including a model of antibiotic-treated mice that mimics what occurs in hospitalized patients (the human population with risk of developing infections with MRE).

- ii) We demonstrate the mechanism of how *Lactobacillus* restricts intestinal MRE colonization, which was not studied by Freter et al (ref. 39, now ref. 40). Specifically, we showed that *Lactobacillus* promotes Clostridiales expansion after antibiotic treatment, which then creates a hostile environment for MRE growth (increase of inhibitory molecules such as butyrate and depletion of specific nutrient sources for MRE growth). This was demonstrated using metagenomic and metabolomic analysis, *in vivo*, *ex vivo* and *in vitro*.
- iii) As opposed to other studies, all our results are derived from the study of multidrug-resistant clinical strains of *Enterobacteriaceae*, rather than the utilization of model bacteria. This is relevant since it is widely accepted that behavior of lab-adapted model strains does not portray fully the behavior of clinical strains. Therefore, it is important to identify mechanisms by which commensal bacteria restrict colonization by those pathogenic strains that are causing clinical problems in susceptible patients, as we do here.
- iv) Most importantly and related to the translation of results into the clinics, we confirmed that the mechanism of cooperation between *Lactobacillus* and Clostridiales to restrict MRE gut colonization is conserved in humans. This was demonstrated by performing multi-omic analysis in hospitalized patients at risk of developing infections with MRE. This is a very relevant and novel result that has not been previously addressed due to the lack of clinical studies analyzing the levels of MRE in combination with metagenomics, metabolomics and proteomics data in hospitalized patients.
- v) Finally, we agree that previous studies using mouse models already showed a role for Clostridiales derived butyrate production in preventing the expansion of *Enterobacteriaceae*. However, this has never been shown in the specific context of this manuscript: administration of *Lactobacillus* and subsequent expansion of Clostridiales after antibiotic treatment. This context is relevant because it mimics what could be happening in hospitalized patients at risk of developing infections with MRE. Moreover, by performing metabolomic and proteomic analysis of the clinical samples, we were able to show that butyrate production by Clostridiales is associated with lower MRE intestinal colonization levels in the human population. A novel and clinically relevant result.

2) Considering that cooperation between indigenous Clostridia and Lactobacilli in conferring colonization resistance and mechanisms by which Clostridia contribute to colonization resistance against E. coli have already been described, the authors would need to break new ground to move the field forward. We suggested that one path forward would be to determine whether Lactobacilli are responsible for enhancing Clostridia recovery after antibiotic treatment (causality) by elucidating the underlying mechanism. The authors argue that these experiments are not necessary. As a result, the study still marks an incremental advance.

We disagree with the reviewer regarding the lack of results demonstrating a causal effect of *Lactobacillus* on Clostridiales expansion and subsequent restriction of MRE gut colonization. In fact, we did show experimentally that *Lactobacillus* enhance Clostridiales recovery:

- i) We showed that *Lactobacillus* administration leads to increase in relative and absolute abundances of Clostridiales in the gut of animals after antibiotic treatment. A result that has been reproduced with two different types of *Lactobacillus* strains (mouse and human origin) and using proper controls (i.e. mice that did not receive *Lactobacillus*).

- ii) Moreover, we showed that increase in *Lactobacillus* abundance in consecutive fecal samples collected from hospitalized patients is significantly associated with an increase in Clostridiales abundance. This last result suggests that *Lactobacillus* contributes to an expansion of bacteria from the order Clostridiales in the human population as well.

We agree with the reviewer that it will be very interesting to find the molecular mechanism by which *Lactobacillus* promotes Clostridiales. However, answering this question requires an extensive set of experiments, both *in vitro*, *in vivo* using mouse models and finally analyzing hospitalized patients to confirm the mechanism in the human population. Nevertheless, we did perform *in vitro* experiments that pinpoint a potential mechanism by which *Lactobacillus* could be promoting the growth of Clostridiales (i.e. by producing specific metabolites that could boost Clostridiales growth):

- i) We analyzed the metabolic profile of *Lactobacillus* supernatants from *in vitro* cultures and detected 3 metabolites that specifically increased in *Lactobacillus* conditioned media- adenine, ribose and lactate.
- ii) We showed that the combination of these 3 metabolites significantly increases the growth rate of a Clostridiales strain *in vitro*.

These results showed a potential mechanism by which *Lactobacillus* could promote Clostridiales recovery. However, as we mentioned previously, further studies, which exceed the scope of this work, should investigate this mechanism both in mice and humans.

In summary, in this manuscript we provide novel results that demonstrate how two major commensals cooperate to restrict gut colonization by life-threatening multidrug-resistant pathogens in both mice and patients. A clinically and scientifically relevant result that highlights the relevance of exploiting microbiome interactions for developing probiotics to treat infections, and that we expect will be of interest for the broad audience of *Nature Communications*.

Reviewer #2 (Remarks to the Author):

I am very impressed by the authors thorough and attentive response to the reviewer comments, complete with extensive analysis of public data. The manuscript was already strong and now has a new level of depth. I really enjoyed seeing how the Lacto OTU was found in other cohorts, and how similar the sequence is to commercial probiotics, for example.

We thank the reviewer for these kind words. We are glad we were able to provide satisfying answers to the reviewer's questions and comments. We also believe this new data will help readers to understand better the implications and relevance of the results described in this study.

My only minor comment is that there are somewhat pervasive and small grammatical errors - the wrong prepositions are used in several places, for example. I advise having a layer of editing for correctness in English.

We followed the reviewer's advice and asked for a native speaker to correct the manuscript.